# Targeting Dermal Fibroblast Senescence: From Cellular Plasticity to Anti-Aging Therapies

**DOI:** 10.3390/biomedicines13081927

**Published:** 2025-08-07

**Authors:** Raluca Jipu, Ionela Lacramioara Serban, Ancuta Goriuc, Alexandru Gabriel Jipu, Ionut Luchian, Carmen Amititeloaie, Claudia Cristina Tarniceriu, Ion Hurjui, Oana Maria Butnaru, Loredana Liliana Hurjui

**Affiliations:** 1Department of Morpho-Functional Sciences II, Physiology Discipline, “Grigore T. Popa” University of Medicine and Pharmacy, 700115 Iași, Romania; raluca.jipu@umfiasi.ro (R.J.); ionela.serban@umfiasi.ro (I.L.S.); loredana.hurjui@umfiasi.ro (L.L.H.); 2Department of General and Oral Biochemistry, Faculty of Dental Medicine, “Grigore T. Popa” University of Medicine and Pharmacy, 700115 Iași, Romania; 3Oral Pathology Department, Faculty of Dental Medicine, “Grigore T. Popa” University of Medicine and Pharmacy, 700115 Iaşi, Romania; jipu.alexandru-gabriel@d.umfiasi.ro (A.G.J.); carmen.amititeloaie@umfiasi.ro (C.A.); 4Department of Periodontology, Faculty of Dental Medicine, “Grigore T. Popa” University of Medicine and Pharmacy, 700115 Iași, Romania; ionut.luchian@umfiasi.ro; 5Department of Morpho-Functional Sciences I, Discipline of Anatomy, “Grigore T. Popa” University of Medicine and Pharmacy, 700115 Iași, Romania; claudia.tarniceriu@umfiasi.ro; 6Hematology Clinic, “Sf. Spiridon” County Clinical Emergency Hospital, 700111 Iași, Romania; 7Biophysics and Medical Physics Department, “Grigore T. Popa” University of Medicine and Pharmacy, 700115 Iași, Romania; ion.hurjui@umfiasi.ro; 8Department Biophysics, Faculty of Dental Medicine, “Grigore T. Popa” University of Medicine and Pharmacy, 700115 Iași, Romania; oana.maria.butnaru@umfiasi.ro; 9Hematology Laboratory, “Sf. Spiridon” County Clinical Emergency Hospital, 700111 Iași, Romania

**Keywords:** dermal fibroblasts, cellular senescence, senolytics, geroprotectors, skin aging

## Abstract

Dermal fibroblasts, the primary stromal cells of the dermis, exhibit remarkable plasticity in response to various stimuli, playing crucial roles in tissue homeostasis, wound healing, and ECM production. This study examines the molecular mechanisms underlying fibroblast plasticity, including key signaling pathways, epigenetic regulation, and microRNA-mediated control. The impact of aging on ECM synthesis and remodeling is discussed, and the diminished production of vital components such as collagen, elastin, and glycosaminoglycans are highlighted, alongside enhanced ECM degradation through upregulated matrix metalloproteinase activity and accumulation of advanced glycation end products. The process of cellular senescence in dermal fibroblasts is explored, with its role in skin aging and its effects on tissue homeostasis and repair capacity being highlighted. The senescence-associated secretory phenotype (SASP) is examined for its contribution to chronic inflammation and ECM disruption. This review also presents therapeutic perspectives, focusing on senolytics and geroprotectors as promising strategies to combat the negative effects of fibroblast senescence. Current challenges in translating preclinical findings to human therapies are addressed, along with future directions for research in this field. This comprehensive review explores the complex interplay between dermal fibroblast plasticity, cellular senescence, and extracellular matrix (ECM) remodeling in the context of skin aging. In conclusion, understanding the complex interplay between dermal fibroblast plasticity, cellular senescence, and extracellular matrix (ECM) remodeling is essential for developing effective anti-aging interventions, which highlights the need for further research into senolytic and geroprotective therapies to enhance skin health and longevity. This approach has shown promising results in preclinical studies, demonstrating improved skin elasticity and reduced signs of aging.

## 1. Introduction

Aging is a complex, multifaceted process characterized by the gradual deterioration of cellular and tissue functions. Cellular senescence, initially defined by Hayflick and colleagues in 1961 as the irreversible slowing of proliferation in normal diploid cells after a finite number of divisions, plays a significant role in this process [1]. This initial observation, which contradicted the theory of unlimited cell growth, opened the path to understanding the link between aging and the limitation of cellular replication capacity.

Senescence and cellular aging, although closely related, are distinct processes. Cellular aging is a gradual and universal process that involves the accumulation of molecular and cellular deficiencies that lead to suboptimal function. In contrast, cellular senescence is a specific response to significant cellular stress or damage, resulting in the irreversible end of the cell cycle and dramatic phenotypic changes [2]. Senescent cells enter a state of permanent cessation, exhibiting characteristics such as resistance to apoptosis, morphological changes, and the secretion of pro-inflammatory factors (senescence-associated secretory phenotype or SASP) [3].

This review focuses on the role of dermal fibroblasts in skin aging and the possible molecular mechanisms involved in this process. Dermal fibroblasts, mesodermal cells essential for maintaining the integrity of connective tissue, exhibit remarkable plasticity and adapt their phenotype and function to diverse conditions. We will analyze the plasticity of dermal fibroblasts, the impact of senescence on their functions, and their contribution to tissue homeostasis and repair in the context of aging. Dermal fibroblasts, the predominant stromal cells of the dermis, play an essential role in maintaining tissue integrity and homeostasis. Unlike epithelial cells, which are polarized and form continuous layers, fibroblasts are mesenchymal cells characterized by remarkable plasticity. This plasticity is manifested by their ability to modify their phenotype and function in response to various internal and external stimuli, allowing them to contribute to a wide range of biological processes, including tissue development, wound healing, and extracellular matrix (ECM) remodeling [4].

Cellular plasticity generally refers to the ability of cells to adapt their phenotype and function based on their microenvironment. For dermal fibroblasts, this translates into a remarkable ability to switch between different functional states, from an inactive state with reduced metabolic activity to activated states characterized by proliferation and increased ECM secretion [5]. This phenotypic flexibility is essential for fibroblasts to adapt to the needs of the tissue and enables them to contribute to diverse biological processes. This plasticity is regulated by a complex array of internal and external signals, including growth factors, cytokines, transcription factors, and epigenetic modifications. It is important to note that dermal fibroblasts are not a homogeneous population, but consist of distinct subpopulations with unique molecular signatures and specialized functions. For example, fibroblasts in the papillary dermis, characterized by the expression of Dkk3 and CD90, differ from those in the reticular dermis, which express higher levels of elastin and fibulin-1, in their expression of ECM components, response to growth factors, and role in wound healing [6]. Furthermore, some fibroblast populations are localized to specific skin layers depending on their cellular origin, such as pre-adipocytes giving rise to fibroblasts after injury [5]. A comprehensive understanding of dermal fibroblast plasticity requires consideration of these specialized subpopulations and their unique characteristics.

This review aims to provide a comprehensive and up-to-date overview of dermal fibroblast plasticity and senescence and their impact on skin aging, with a secondary goal of highlighting potential therapeutic targets for combating age-related skin changes. To achieve this, we examine the molecular mechanisms underlying dermal fibroblast plasticity, discuss the impact of aging on ECM synthesis and remodeling, explore the role of cellular senescence in dermal fibroblasts, and present therapeutic perspectives that focus on senolytics and geroprotectors. This review is intended for researchers in the fields of dermatology, cell biology, aging research, and regenerative medicine who are seeking an overview of the topic.

## 2. Molecular Mechanisms of Dermal Fibroblast Plasticity

The ability of dermal fibroblasts to adapt and change their behavior (what we call plasticity) is carefully controlled by a complex network of signaling pathways and epigenetic modifications. To truly understand the diverse roles that fibroblasts play in maintaining healthy skin, healing wounds, and the aging process, it is crucial to understand these underlying mechanisms.

### 2.1. Key Signaling Pathways

Several key signaling pathways control fibroblast behavior and plasticity. These pathways are often connected and influence each other, creating a complex regulatory network. These key signaling pathways play essential roles in regulating fibroblast behavior and plasticity, contributing to tissue homeostasis and repair in normal conditions. However, dysregulation of these pathways can lead to a variety of pathological conditions, which highlights the importance of understanding their dual roles in both health and disease. These pathways are often interconnected and influence each other, creating a complex regulatory network.


*Transforming Growth Factor-beta (TGF-β) Signaling*


TGF-β is a pleiotropic cytokine with diverse effects on fibroblasts. In physiological conditions, TGF-β plays a key role in regulating extracellular matrix (ECM) production, cell proliferation, and differentiation [7], ensuring proper tissue structure and repair. TGF-β signaling activates SMAD proteins, which leads to transcriptional regulation of genes involved in ECM synthesis (e.g., collagen, fibronectin) and other fibroblast functions [8]. However, dysregulation of TGF-β signaling has been linked to fibrosis and other pathological conditions. Excessive activation of TGF-β can lead to overproduction of ECM components, resulting in tissue fibrosis in various organs, including the skin. This dysregulation can be triggered by chronic inflammation, genetic factors, or other environmental stimuli [9].


*Fibroblast Growth Factor (FGF) Signaling*


FGFs are a family of growth factors that bind to FGF receptors and activate downstream signaling cascades that involve MAPK/ERK and PI3K/Akt pathways [10]. Under normal conditions, FGF signaling pathways influence cell proliferation, migration, and differentiation, which is essential for wound healing and tissue repair. However, in pathological conditions, sustained FGF signaling can contribute to tumor growth and angiogenesis [11]. In the context of fibrosis, excessive FGF signaling can promote fibroblast proliferation and ECM deposition, thus contributing to tissue scarring.


*Platelet-Derived Growth Factor (PDGF) Signaling*


PDGF is a potent mitogen for fibroblasts, stimulating cell proliferation and migration [12]. PDGF signaling involves the activation of receptor tyrosine kinases and downstream signaling through PI3K/Akt and MAPK/ERK pathways [13]. PDGF plays a critical role in wound healing and tissue regeneration [14], promoting fibroblast recruitment and proliferation at the site of injury. In pathological conditions, sustained or excessive PDGF signaling can contribute to conditions such as hypertrophic scarring and fibrosis. Overexpression of PDGF can lead to excessive fibroblast proliferation and ECM deposition, which disrupts normal tissue architecture [15].


*Wnt Signaling*


The Wnt signaling pathway is involved in regulating various aspects of fibroblast behavior, including proliferation, differentiation, and ECM production [16]. In healthy skin, Wnt signaling helps maintain proper cell differentiation, tissue architecture, and ECM production. Disruptions in Wnt signaling have been implicated in various skin diseases and aging processes. Abnormal Wnt signaling can lead to dysregulated fibroblast functions and thus contributes to skin conditions such as fibrosis, psoriasis, and atopic dermatitis by affecting collagen deposition and ECM remodeling in aging. As organisms age, Wnt signaling can become dysregulated, impacting fibroblast functionality and leading to skin aging characteristics, such as reduced elasticity, increased sagging, and a diminished capacity for wound healing.

### 2.2. Epigenetic Regulation

Epigenetic modifications, such as DNA methylation and histone modifications, play a significant role in regulating the gene expression in fibroblasts and, consequently, their phenotype. These are not actual changes in the DNA sequence itself, but rather epigenetic modifications that affect how the DNA is packaged and how genes are expressed. Epigenetic modifications can dynamically influence fibroblast behavior by altering gene expression patterns that establish proliferation, differentiation, and response to environmental signals, thereby affecting healing and tissue regeneration [17]. Because epigenetic changes are reversible, targeting these modifications through pharmacological agents or lifestyle interventions presents a promising therapeutic approach for increasing the functions of fibroblasts in various diseases, including fibrosis and other skin disorders [18].


*DNA Methylation*


DNA methylation, an epigenetic modification that alters gene expression without changing the DNA sequence, plays a crucial role in fibroblast differentiation and function. By affecting the expression of genes involved in collagen production, extracellular matrix composition, and cell proliferation, methylation patterns strongly influence fibroblast behavior. Key genes such as *COL1A1, ACTA2*, and *PDGFRA* are regulated through methylation, affecting cellular activities and tissue function [19]. Abnormal DNA methylation in fibroblasts has been associated with various pathological conditions, including fibrosis, impaired wound healing, and certain cancers. Understanding these methylation patterns offers promising therapeutic potential, such as the use of DNA methyltransferase inhibitors to modify fibroblast behavior in disease states.


*Histone Modifications*


Histone modifications, such as acetylation and methylation, can alter the chromatin structure and gene expression. These modifications can influence fibroblasts’ response to various stimuli. Histone modifications, including acetylation and methylation, are crucial epigenetic mechanisms that play a significant role in regulating gene expression and chromatin structure in fibroblasts. These modifications can dynamically alter the accessibility of DNA to transcription factors and other regulatory proteins, thereby influencing the fibroblast’s response to various stimuli. Acetylation of histones generally promotes gene activation by loosening the chromatin structure, while methylation can either activate or repress genes depending on the specific residues that are modified and the degree of methylation [20]. In fibroblasts, these histone modifications can affect key cellular processes such as differentiation, proliferation, and extracellular matrix production. Understanding the intricate patterns of histone modifications in fibroblasts provides insights into their behavior in both normal physiological conditions and disease states, potentially offering new targets for therapeutic interventions in fibrosis-related disorders and wound healing [21].

### 2.3. MicroRNAs (miRNAs)

MicroRNAs are small non-coding RNAs that regulate gene expression post-transcriptionally. They can target mRNAs involved in cell cycle progression, differentiation, and ECM production, thereby impacting fibroblast plasticity. MicroRNAs (miRNAs) are short, non-coding RNA molecules that play a crucial role in regulating gene expression at the post-transcriptional level in fibroblasts and other cell types. These small RNAs, typically 20–25 nucleotides in length, function by binding to complementary sequences on target messenger RNAs (mRNAs), which leads to their degradation or translational repression. In fibroblasts, miRNAs can significantly influence cellular plasticity by modulating the expression of genes involved in key processes such as cell cycle progression, differentiation, and extracellular matrix (ECM) production [22]. By adjusting these fundamental cellular functions, miRNAs contribute to the dynamic nature of fibroblasts, allowing them to respond and adapt to various environmental signals and physiological demands. Understanding the complex network of miRNA-mediated regulation in fibroblasts provides valuable insights into their behavior in normal tissue homeostasis, wound healing, and pathological conditions like fibrosis and cancer [23]. For example, the *miR-29* family (including *miR-29a, miR-29b*, and *miR-29c*) has been shown to play a critical role in regulating collagen expression. These miRNAs directly target mRNAs that encode collagen types I and III, and their downregulation in fibrotic conditions leads to increased collagen synthesis and ECM deposition [24]. Conversely, *miR-21* is often upregulated in fibrotic environments, which promotes fibroblast proliferation and migration due to *miR-21* targeting genes involved in apoptosis and cell cycle regulation, such as PTEN and PDCD4 [25]. Furthermore, *miR-155* has been implicated in the regulation of inflammation and ECM remodeling. Upregulation of *miR-155* in fibroblasts can promote the expression of pro-inflammatory cytokines and MMPs, contributing to ECM degradation and tissue damage [26]. Finally, *miR-196a* has been shown to regulate collagen expression and fibrosis by targeting genes involved in the TGF-β signaling pathway [27]. These are just a few examples of the many miRNAs that contribute to fibroblast plasticity, which highlights the complex and multifaceted nature of this regulatory network.

Fibroblasts are essential cells involved in building and remodeling the extracellular matrix (ECM), which is vital for keeping tissues structured and functioning properly. The ECM maintained maintained in a dynamic balance between synthesis and degradation, and fibroblasts play a crucial role in maintaining this equilibrium. These cells are the first to produce ECM components and are responsible for constantly turning over and adjusting the matrix as needed, whether through normal activity or in response to various physiological and pathological stimuli. Fibroblasts synthesize a large number of ECM proteins, including collagens, elastin, fibronectin, and proteoglycans, which come together to form the supportive framework [28]. In addition to manufacturing these structural proteins, fibroblasts also produce enzymes called matrix metalloproteinases (MMPs), which are responsible for breaking down components of the ECM. MMPs are a family of zinc-dependent endopeptidases that degrade various components of the ECM. Their activity is carefully controlled by tissue inhibitors of metalloproteinases (TIMPs), which bind to MMPs and prevent excessive degradation. TIMPs inhibit MMP activity by binding to their active sites, which prevents them from interacting with ECM substrates. The balance between ECM creation and breakdown is tightly managed by fibroblasts, which allows tissues to stay healthy and adapt to different conditions [29]. This balance is dynamic, allowing for ECM remodeling during tissue development, wound healing, and normal tissue turnover. TIMPs are a family of proteins that bind to MMPs and inhibit their activity, which maintains ECM homeostasis [30]. However, this balance can be disrupted in various pathological conditions. For example, in skin aging, increased MMP activity and decreased TIMP expression contribute to the degradation of collagen and elastin fibers, leading to wrinkle formation and loss of skin elasticity. The balance between ECM synthesis and degradation is tightly controlled by fibroblasts, which allows for the maintenance of tissue homeostasis and adaptation to changing environmental conditions. In response to injury or disease, fibroblasts can become activated, which leads to increased ECM production and remodeling. This process is critical for wound healing and tissue repair but can also contribute to fibrosis and scarring if dysregulated [31]. Fibroblasts also interact with other cell types, such as immune cells and endothelial cells, through the secretion of growth factors and cytokines, further influencing the ECM composition and tissue microenvironment. The ability of fibroblasts to sense and respond to mechanical forces through mechanotransduction pathways allows them to adapt ECM synthesis and remodeling to the mechanical needs of the tissue. Recent studies have shown that fibroblast populations are diverse within and between different tissues, which suggests that certain subsets may have specialized roles in regulating the ECM [32]. Understanding the complex relationship between fibroblasts and the ECM is essential for developing therapeutic strategies that target fibrosis, tissue engineering, and regenerative medicine applications.

### 2.4. Signaling Pathways and Dermal Fibroblast Subpopulations

While the signaling pathways described above (TGF-β, PDGF, Wnt, FGF) are relevant to all dermal fibroblasts, it is increasingly evident that different subpopulations may exhibit distinct responses to these signals. For instance, papillary fibroblasts, being more responsive to Wnt signaling, have been shown to exhibit increased proliferation and migration in response to Wnt ligands compared to reticular fibroblasts (Figure 1) [33]. Reticular fibroblasts, in contrast, may be more sensitive to TGF-β, leading to increased collagen production and a greater propensity for scar formation [34]. Furthermore, the expression of specific signaling receptors, such as the PDGF receptor alpha (PDGFRA), may vary among fibroblast subpopulations, which leads to differences in their responses to PDGF [35]. Further research is needed to fully elucidate the specific signaling mechanisms that regulate the unique functions of different dermal fibroblast subpopulations.

Wnt: wingless/integrase-1 (refers to a family of signaling pathways);GSK: glycogen synthase Kkinase;CBF-1: CBF1 transactivator/also known as RBP-Jκ (recombination signal binding protein for immunoglobulin kappa J region);APC: adenomatous polyposis coli;β-catenin: beta-catenin protein;TCF: T-cell factor;LEF: lymphoid enhancer-binding factor;CSL: CBF1/suppressor of hairless/LAG-1 (a DNA-binding protein involved in Notch signaling).

Reticular fibroblasts:Raf: rapidly accelerated fibrosarcoma kinase;MEK1/2: mitogen-activated protein kinase kinase 1/2;ERK1/2: extracellular signal-regulated kinase 1/2;RBP-J: recombination signal binding protein for immunoglobulin kappa J region (also known as CBF1);Dll: delta-like ligand (e.g., Dll1, Dll3, Dll4);Notch: refers to the Notch receptor signaling pathway;Smad2, Smad3, Smad4: mothers against decapentaplegic homologs 2, 3, and 4 (intracellular signaling mediators for TGF-β).

## 3. Extracellular Matrix (ECM) and the Role of Fibroblasts

### 3.1. Extracellular Matrix

The ECM is a complex and dynamic network of macromolecules that surrounds cells in connective tissue (Figure 2). This complex network is not just a passive support, it plays an essential role in regulating cellular functions, including proliferation, differentiation, migration, the cell shape, gene expression, and cell survival. The ECM acts as a reservoir of growth factors, cytokines, and matricellular proteins, which are secreted and stored within the matrix, allowing for controlled release and presentation to cells and directly influencing cell behavior through interactions with cell surface receptors, such as integrins, discoidin domain receptors [36,37] (DDRs), and syndecans. Integrins, for example, are transmembrane receptors that mediate cell–ECM adhesion and transmit bidirectional signals between the cell and its environment, influencing intracellular signaling pathways like MAPK and Rho GTPases, which regulate cell motility and cytoskeletal organization [38]. Matricellular proteins like tenascins and osteopontin modulate cell–ECM interactions without directly contributing to the structural integrity of the matrix [36].

The composition of the ECM is extremely varied and specific to each type of tissue, which reflects its specialized functions and the dynamic requirements of the resident cells. In the dermis, the main components include the following:*Collagen*

Collagen is the major structural component of the dermal ECM, representing approximately 70–80% of the skin’s dry weight and providing the skin with its tensile strength and structural integrity [39]. Collagen is synthesized as a procollagen precursor, undergoing post-translational modifications, including hydroxylation of proline and lysine residues (requiring vitamin C as a cofactor) and glycosylation, before being secreted into the extracellular space where it is cleaved by procollagen peptidases to form mature collagen molecules that self-assemble into fibrils and fibers. There are at least 28 different types of collagen, each of which is encoded by a distinct gene. Type I collagen is predominant in the reticular dermis, provides resistance to stretching, and is synthesized mainly by dermal fibroblasts. Other important collagen types include type III collagen, which is more abundant in the papillary dermis. It plays a role in the early stages of wound healing by providing a more pliable matrix that facilitates cell migration [40]. Type IV collagen is a major component of the basement membrane underlying the epidermis and surrounding blood vessels. Type VII collagen forms anchoring fibrils that attach the basement membrane to the underlying dermis, ensuring epidermal–dermal cohesion [41]. Fibrillar collagens (types I, II, III, V, and XI) form the major structural framework of the ECM, while non-fibrillar collagens (e.g., types IV, VI, VII) have specialized functions in basement membranes, anchoring fibrils, and pericellular matrices. Collagen synthesis is tightly regulated by a variety of growth factors, including TGF-β, connective tissue growth factor (CTGF), and platelet-derived growth factor (PDGF), as well as mechanical cues and inflammatory mediators.


*Elastin*


Elastin provides elasticity and the ability of connective tissue to stretch and recoil. Elastin fibers form a three-dimensional network [42] that allows the skin to deform reversibly and maintain tissue resilience. Elastin is synthesized primarily by fibroblasts in the dermis, as well as by smooth muscle cells in the walls of blood vessels, during development and early adulthood; elastin synthesis declines significantly with age [43]. Elastin monomers (tropoelastin) are secreted and assembled onto microfibrils composed of fibrillin-1, fibulin-5, and other proteins, which forms elastic fibers. The cross-linking of tropoelastin molecules by lysyl oxidase (LOX) is essential for the formation of mature, functional elastin fibers [44]. The degradation of elastin, caused by factors such as chronic exposure to ultraviolet (UV) radiation from the sun and increased levels of proteolytic enzymes (matrix metalloproteinases—MMPs, particularly MMP-2 and MMP-9, and elastases) released by inflammatory cells like neutrophils and macrophages, contributes to the loss of skin elasticity (elastosis) and the formation of wrinkles associated with chronological aging and photoaging. Advanced glycation end-products (AGEs), formed through the non-enzymatic glycation of proteins, also contribute to elastin stiffening and reduced elasticity in aged skin [45].


*Proteoglycans*


Proteoglycans are large macromolecules composed of a core protein covalently linked to glycosaminoglycan (GAG) chains. GAGs attract and retain water molecules, contributing to ECM hydration and compressibility [46]. These GAGs, which are negatively charged due to the presence of sulfate and carboxyl groups, attract and retain water molecules, contributing to the hydration, turgor, and compressibility of the ECM [47]. Proteoglycans play a critical role in regulating ECM assembly, cell adhesion, growth factor signaling, and tissue homeostasis. Decorin, a small leucine-rich proteoglycan (SLRP), binds to collagen fibrils and regulates their diameter and organization, inhibiting excessive collagen deposition and fibrosis. Biglycan, another SLRP, also interacts with collagen and modulates cell signaling through Toll-like receptors (TLRs). Versican, a large aggregating proteoglycan, contributes to tissue hydration and cell migration during development and wound healing [48].


*Glycosaminoglycans (GAGs)*


GAGs are long, unbranched, linear polysaccharides composed of repeating disaccharide units. GAGs are highly negatively charged due to the presence of sulfate and uronic acid groups, and they have a high affinity for water. Major GAGs in the skin include hyaluronic acid (hyaluronan), chondroitin sulfate, dermatan sulfate, heparan sulfate, and keratan sulfate [49]. Hyaluronic acid (HA) is a non-sulfated GAG with an exceptional ability to bind and retain water, up to 1000 times its weight, and thus significantly contributes to the hydration, viscoelasticity, compressibility, and mechanical properties of the dermal ECM. HA also plays a crucial role in cell migration, proliferation, and inflammation by interacting with cell surface receptors such as CD44 and receptor for hyaluronan-mediated motility (RHAMM). Chondroitin sulfate and dermatan sulfate are often found covalently attached to proteoglycans, influencing their interactions with other ECM components and regulating cell signaling [50]. GAGs, through their interaction with proteins and growth factors, modulate their activity and influence key cellular processes, including cell adhesion, migration, proliferation, and differentiation.


*Adhesive Glycoproteins*


Adhesive glycoproteins are a diverse group of multi-domain proteins that mediate cell–ECM and cell–cell interactions, playing a crucial role in cell adhesion, migration, differentiation, and tissue organization [51]. They are crucial for cell adhesion, migration, and signaling. Fibronectin is a large, multifunctional glycoprotein that binds to a variety of ECM components, including collagen, fibrin, heparin, and heparan sulfate, as well as to cell surface integrin receptors, facilitating cell adhesion, migration, wound healing, and tissue remodeling. Fibronectin exists as a soluble dimer in plasma and as an insoluble multimer in the ECM, with different isoforms being generated by alternative splicing the fibronectin gene. Laminins are a family of heterotrimeric glycoproteins that are major components of basement membranes, playing a crucial role in the adhesion, migration, and differentiation of epithelial and endothelial cells, as well as in the structural organization and stability of the basement membrane. Laminins bind to cell surface integrins, dystroglycan, and other receptors, influencing cell signaling and tissue morphogenesis [52]. Other important adhesive glycoproteins include tenascins, thrombospondins, and vitronectin, each of which has distinct roles in regulating cell–ECM interactions and tissue remodeling.

The ECM plays a vital role in maintaining the homeostasis and functions of connective tissue, including the following:Structural support: provides structural support and organization of tissues;Segregation: separates different tissue compartments, creating specific microenvironments for different cell types;Cellular signaling: participates in cell signaling, influencing cell proliferation, differentiation, and survival;Cell migration: facilitates cell migration, which is essential in developmental processes, wound healing, and immune responses;Tissue homeostasis: regulates tissue growth, remodeling, and repair processes [53].

### 3.2. Role of Fibroblasts in ECM Synthesis and Remodeling

Fibroblasts are fundamental cells for tissue architecture and functionality, with the primary role of continuously constructing and remodeling the extracellular matrix (ECM). Within the context of ECM synthesis and remodeling, fibroblasts actively manage the dynamic equilibrium between the production and degradation of matrix components, ensuring optimal tissue functionality. These cells are the first to produce ECM components and are responsible for constantly turning over and adjusting the matrix as needed, whether through normal activity or in response to various physiological and pathological stimuli. Fibroblasts synthesize a large number of ECM proteins, including collagens, elastin, fibronectin, and proteoglycans, which come together to form the supportive framework of [54]. In addition to manufacturing these structural proteins, fibroblasts also produce enzymes called matrix metalloproteinases (MMPs), which are responsible for breaking down components of the ECM. MMPs are a family of zinc-dependent endopeptidases that degrade various components of the ECM. Their activity is carefully controlled by tissue inhibitors of metalloproteinases (TIMPs), which bind to MMPs and prevent excessive degradation. TIMPs inhibit MMP activity by binding to their active sites, preventing them from interacting with ECM substrates This ensures that ECM degradation is tightly controlled, preventing excessive breakdown of the matrix.

The balance between ECM creation and breakdown is tightly managed by fibroblasts, allowing tissues to stay healthy and adapt to different conditions [55]. MMP activity is finely controlled by tissue inhibitors of metalloproteinases (TIMPs), a family of proteins that bind to MMPs and inhibit their activity, maintaining ECM homeostasis [56]. This dynamic equilibrium between MMPs and TIMPs is crucial for preserving tissue structure and function However, this balance can be disrupted in various pathological conditions. For example, in skin aging, increased MMP activity and decreased TIMP expression contribute to the degradation of collagen and elastin fibers, leading to wrinkle formation and loss of skin elasticity. The balance between ECM synthesis and degradation is tightly controlled by fibroblasts, allowing for the maintenance of tissue homeostasis and adaptation to changing environmental conditions. In response to injury or disease, fibroblasts can become activated, leading to increased ECM production and remodeling.

This process is critical for wound healing and tissue repair but can also contribute to fibrosis and scarring if dysregulated [57]. Fibroblasts also interact with other cell types, such as immune cells and endothelial cells, through the secretion of growth factors and cytokines, further influencing the ECM composition and tissue microenvironment. The ability of fibroblasts to sense and respond to mechanical forces through mechanotransduction pathways allows them to adapt ECM synthesis and remodeling to the mechanical needs of the tissue. Recent studies have shown that fibroblast populations are diverse within and between different tissues, suggesting that certain subsets may have specialized roles in regulating the ECM [58]. Understanding the complex relationship between fibroblasts and the ECM is essential for developing therapeutic strategies targeting fibrosis, tissue engineering, and regenerative medicine applications. Therefore, maintaining the appropriate MMP/TIMP regulatory balance is essential for ensuring proper ECM turnover, tissue homeostasis, and effective responses to injury or disease, which makes it a critical target for therapeutic interventions aimed at modulating tissue remodeling.

### 3.3. ECM Composition and Dermal Fibroblast Subpopulations

The ECM is not only a product of fibroblast activity, but also a key regulator of fibroblast phenotype and function. Distinct dermal fibroblast subpopulations contribute differently to the composition and organization of the ECM. Papillary fibroblasts, characterized by their stellate morphology and proximity to the epidermis, are primarily responsible for synthesizing type III collagen and components of the basement membrane, such as laminin and collagen IV, and thus contribute to the structural integrity of the epidermal–dermal junction and provide support for keratinocytes [59]. Reticular fibroblasts, which are larger and more elongated, are the main producers of type I collagen, providing tensile strength to the deeper layers of the dermis and organizing the collagen fibers into thick bundles [60]. Furthermore, the specific ECM microenvironment surrounding different fibroblast subpopulations, such as the presence of hyaluronic acid in the papillary dermis, can influence their response to growth factors and cytokines, which creates a feedback loop that further shapes their phenotype and function. In addition, following injury, pre-adipocytes transdifferentiate to fibroblasts and deposit ECM at the injury site (Figure 3) [61].

## 4. Impact of Aging on Extracellular Matrix Synthesis and Remodeling: A Molecular and Cellular Perspective

Aging processes, over time, have a great impact on the structure and function of the dermal ECM, contributing significantly to the characteristic changes observed in aged skin. These changes are not just due to a simple decline in cell activity; instead, they result from a complex mix of molecular and cellular processes that gradually break down and weaken the ECM, which leads to aged skin’s characteristic look and feel [62]. These processes include changes in gene expression, protein synthesis, enzymatic activity, and cell–cell communication, all of which contribute to a decline in the structural integrity and functional capacity of the dermis. Furthermore, environmental factors such as UV radiation, pollution, and smoking can accelerate these age-related changes, leading to premature skin aging [63].

### 4.1. Diminished ECM Production

A key feature of skin aging is the decreased production of important components of the extracellular matrix. This decline impacts several essential structural proteins and glycosaminoglycans, which are vital for maintaining healthy, youthful skin:*Collagen-decreased synthesis rate and altered composition*

The synthesis of collagen, especially type I collagen, undergoes a significant age-related decline [64]. This decline happens because dermal fibroblasts have decreased collagen gene transcription [65] and impaired post-translational modifications, including decreased hydroxylation and glycosylation [66]. Specifically, the activity of prolyl hydroxylase, an enzyme that is responsible for hydroxylating proline and lysine residues in collagen, decreases with age, which results in reduced collagen stability and cross-linking [67]. The ratio of type I/type III collagen also changes with age, with a relative increase in type III collagen, which has a smaller fiber diameter and lower tensile strength [68]. The expression of collagen chaperones, such as heat shock protein 47 (HSP47), which are essential for proper collagen folding and secretion, is reduced in aged fibroblasts [69]. Meanwhile, existing collagen fibers exhibit increased cross-linking, which results in a less organized and more rigid structure with reduced tensile strength. These cross-links are formed by advanced glycation end products (AGEs) and enzymatic cross-linking mediated by lysyl oxidase (LOX) [70]. The accumulation of AGEs not only stiffens collagen fibers but also makes them more susceptible to degradation by matrix metalloproteinases (MMPs) [71]. This altered collagen structure contributes significantly to the reduced elasticity and increased susceptibility to wrinkles and sagging observed in aged skin.


*Elastin: breakdown and loss of elasticity*


Elastin fibers, responsible for the elasticity and resilience of the dermis, also undergo significant age-related changes. The synthesis of tropoelastin, the precursor to elastin, diminishes with age, which results in fewer elastin fibers and reduced overall elastic recoil [72]. The decline in tropoelastin synthesis is associated with decreased expression of the elastin gene and reduced stability of elastin mRNA [73]. In addition, age-related changes in the microfibrillar network, which provides a scaffold for elastin deposition, can impair the proper assembly and organization of elastin fibers [74]. Furthermore, existing elastin fibers become fragmented and cross-linked, which leads to reduced elasticity and increased susceptibility to wrinkle formation [75]. Elastin fragmentation is primarily mediated by MMPs, particularly MMP-2, MMP-9, and MMP-12, which are upregulated in aged skin [76]. Cross-linking of elastin fibers by AGEs and LOX further reduces their elasticity and makes them more resistant to degradation [77]. The accumulation of damaged elastin fragments in the dermis, known as solar elastosis, is a hallmark of photoaged skin [78]. This loss of elasticity contributes to the loss of skin turgor and the appearance of wrinkles and sagging.


*Glycosaminoglycans (GAGs)–reduced hydration and modified biomechanical properties*


GAGs, including hyaluronic acid, are essential for maintaining the hydration and viscoelasticity of the dermis [79]. Age-related declines in GAG synthesis lead to decreased water retention in the dermis, contributing to increased skin dryness and altered biomechanical properties. The synthesis of hyaluronic acid (HA) by HA synthases (HAS1, HAS2, and HAS3) is reduced in aged fibroblasts, which leads to a decrease in HA content in the dermis [80]. In addition, the degradation of HA by hyaluronidases (HYAL1, HYAL2, and HYAL3) is increased with age, further contributing to the decline in HA levels [81]. The sulfation of other GAGs, such as chondroitin sulfate and dermatan sulfate, is also reduced in aged skin, which affects their interactions with other ECM components and their ability to regulate cell signaling [82]. This diminished hydration further contributes to the appearance of wrinkles and reduced skin resilience [83].

### 4.2. Enhanced ECM Degradation

The age-related decline in ECM production is further exacerbated by increased ECM degradation, which is mediated by several key processes:Upregulation of matrix metalloproteinases (MMPs): Various MMPs, especially MMP-1, MMP-2, and MMP-9, become more active with age [84]. The expression of MMPs is regulated by a variety of factors, including growth factors, cytokines, and UV radiation [85]. Aged fibroblasts exhibit increased expression of MMPs due to increased activity of transcription factors such as AP-1 and NF-κB [86]. Furthermore, the levels of reactive oxygen species (ROS) are elevated in aged skin, which can activate MMPs and promote ECM degradation [87]. This increased MMP activity, often exceeding the capacity of tissue inhibitors of metalloproteinases (TIMPs) to neutralize them, leads to the net degradation of collagen and elastin fibers [88]. The imbalance between MMPs and TIMPs is a key factor in age-related ECM degradation. The expression of TIMPs, particularly TIMP-1 and TIMP-2, is reduced in aged fibroblasts, which further contributes to the increased MMP activity [89]. The imbalance between MMP and TIMP activity makes the ECM structure less stable and accelerates aging phenotypes;Accumulation of advanced glycation end products (AGEs): AGEs are formed through the non-enzymatic glycation of ECM proteins, particularly collagen and elastin. AGEs cross-link ECM molecules, increasing their rigidity and susceptibility to degradation [90]. Glycation is a process in which reducing sugars, such as glucose and fructose, react with amino groups in proteins to form Schiff bases, which undergo further reactions to form irreversible AGEs. AGEs accumulate in the skin with age, particularly in long-lived proteins such as collagen and elastin [91]. Furthermore, AGEs stimulate the production of pro-inflammatory cytokines and reactive oxygen species (ROS), further damaging the ECM;Chronic low-grade inflammation: Chronic low-grade inflammation is a hallmark of aging and significantly contributes to ECM degradation. Inflammatory mediators, such as cytokines and chemokines, promote the activity of MMPs and inhibit collagen synthesis [84,92]. Inflammation is characterized by elevated levels of pro-inflammatory cytokines, such as TNF-α, IL-6, and IL-1β, in the circulation and in tissues [93,94]. This combination further accelerates the breakdown of the ECM.

### 4.3. Cellular and Microenvironmental Alterations

Age-related changes in the cellular microenvironment also play a main role in altered ECM synthesis and remodeling:*Senescent Fibroblasts and the Senescence-Associated Secretory Phenotype (SASP)*

The accumulation of senescent fibroblasts in the dermis is a key feature of aging. Cellular senescence is a state of irreversible growth arrest characterized by morphological and functional changes. Senescent cells accumulate in tissues with age due to DNA damage, telomere shortening, oxidative stress, and oncogene activation [95]. These senescent fibroblasts release a complex mixture of factors, known as the SASP, which includes pro-inflammatory cytokines, growth factors, and MMPs. The SASP is a complex and dynamic secretome that includes a variety of factors, such as IL-6, IL-8, MMP-1, MMP-3, and VEGF. The SASP contributes to chronic inflammation and further promotes ECM degradation [96]. The SASP can also induce senescence in neighboring cells, which leads to a self-perpetuating cycle of senescence and inflammation [95].


*Impaired Cell–Cell Interactions*


The interactions between fibroblasts and other dermal cells, such as keratinocytes, become less efficient with age. Fibroblasts and keratinocytes communicate with each other through direct cell–cell contact and through the release of soluble factors, such as growth factors and cytokines. These altered interactions may result in impaired communication and coordination of ECM production and remodeling [97]. Age-related changes in the expression of cell adhesion molecules, such as integrins and cadherins, can disrupt cell–cell interactions and impair the ability of fibroblasts and keratinocytes to communicate with each other. Furthermore, the decline in growth factor signaling in aged skin can impair the ability of keratinocytes to stimulate collagen synthesis in fibroblasts [98].


*Altered Growth Factor Signaling*


The responsiveness of fibroblasts to various growth factors, such as TGF-β and FGF, is changed with age [99]. Growth factors play a critical role in regulating ECM synthesis and remodeling. TGF-β stimulates collagen synthesis and inhibits MMP expression in fibroblasts, while FGF stimulates fibroblast proliferation and migration [98]. These changes reduce the ability of fibroblasts to effectively produce and remodel the ECM.

### 4.4. Clinical Consequences

The age-related changes in the ECM’s structure and function have several clinical consequences, including:*Increased wrinkle formation and reduced skin elasticity*: The elastin content, coupled with increased ECM degradation, directly leads to visible signs of aging such as fine lines and deeper wrinkles [100]. The loss of elasticity reduces the skin’s ability to recoil after stretching, leading to persistent wrinkles and disorganization of collagen fibers, which in turn disrupt the smooth structure of the skin and contribute to textural irregularities [101];*Skin sagging and loss of turgor*: The reduced collagen and elastin support results in a loss of skin volume and a decrease in the skin’s ability to resist gravity. This leads to sagging, particularly in areas like the cheeks, jawline, and under the eyes. The decreased turgor, or skin fullness, makes the skin appear thinner and more fragile [102]. Changes in the subcutaneous fat distribution, which also occur with aging, further contribute to the loss of facial volume and sagging;*Increased skin dryness and roughness*: The decline in GAGs, especially hyaluronic acid, reduces the skin’s ability to retain moisture, leading to increased dryness and a rough, uneven texture. This dryness can exacerbate the appearance of wrinkles and fine lines and can compromise the skin’s barrier function, making it more susceptible to irritants and allergens. The altered lipid composition of the stratum corneum, which also occurs with aging, further contributes to the increased skin dryness;*Impaired wound healing*: The age-related decline in fibroblast function and ECM remodeling capacity impairs the skin’s ability to heal wounds effectively. Aged fibroblasts exhibit reduced proliferation and migration, and their ability to synthesize new collagen and other ECM components is compromised. The increased levels of MMPs in aged skin can also disrupt the formation of a stable wound matrix, leading to delayed wound closure and increased risk of scarring [103]. Furthermore, the reduced vascularity in aged skin can impair oxygen and nutrient delivery to the wound site, further delaying the healing process;*Increased susceptibility to skin damage and infections:* The thinner, less elastic, and more fragile skin is more vulnerable to damage from external factors such as UV radiation, mechanical trauma, and chemical irritants [102]. The compromised skin barrier function makes it easier for pathogens to penetrate the skin, which increases the risk of infections. The reduced immune function in aged skin further contributes to an increased susceptibility to infections and delayed wound healing [104].

Aging significantly affects ECM synthesis and remodeling. Collagen and elastin synthesis decreases, while MMP activity increases, which leads to increased ECM degradation. This imbalance contributes to the loss of skin elasticity and firmness, wrinkle formation, and other signs of aging. ECM signaling and cell adhesion functions are also compromised [105].

## 5. Senescence of Dermal Fibroblasts

Senescence of dermal fibroblasts is a complex, multifactorial process that significantly contributes to skin aging and profoundly impacts tissue homeostasis and wound repair capacity. This process, initiated by various factors inducing cellular stress, is characterized by irreversible cell cycle arrest and distinct phenotypic changes, with profound implications for the tissue microenvironment [106]. Factors that trigger fibroblast senescence include telomere shortening, replicative stress, oxidative stress, DNA damage, and mitochondrial dysfunction (Figure 4) [107]. In recent years, studies have proved the involvement of oxidative stress in different pathologies, like cardiovascular, kidney, neurodegenerative, pulmonary, and malignant diseases, as well as in the process of aging [108].

### 5.1. Senescent Cell Secretome (SASP)

Senescent cells are not passive. They secrete a complex mixture of bioactive molecules known as the senescence-associated secretory phenotype (SASP). The SASP includes pro-inflammatory cytokines (IL-6, IL-1β, TNF-α), chemokines (CCL2, CXCL8), matrix metalloproteinases (MMPs), growth factors (VEGF, TGF-β), and other molecules. The SASP’s composition and intensity vary depending on the cell type, senescence status, and microenvironmental context. The SASP contributes significantly to the negative impact of senescence on connective tissue, amplifying chronic inflammation, disrupting ECM remodeling, and impairing wound repair [96].

### 5.2. Therapeutic Perspectives: Senolytics and Geroprotectors

Combating the negative effects of fibroblast senescence is a major goal of current research. Two promising therapeutic strategies are senolytics and geroprotectors. Senolytics selectively eliminate senescent cells, while geroprotectors slow the aging process by acting on multiple mechanisms.

Senolytics are a class of compounds designed to selectively induce apoptosis in senescent cells. The molecular basis for their action consists in exploiting the upregulation of pro-survival pathways in senescent cells, such as the Bcl-2/Bcl-xL anti-apoptotic proteins and the p53/p21/serpine pathway [109]. The efficacy of senolytics has been demonstrated in various preclinical models. For instance, Xu et al. (2018) showed that intermittent treatment with dasatinib and quercetin in aged mice led to improved physical function, reduced senescence markers, and an extended healthspan [110]. Also, Yousefzadeh et al. (2018) demonstrated that fisetin reduced senescence markers in multiple tissues and extended the median and maximum lifespan in mice [111].

The mechanism of action of senolytics involves the following:Selective induction of apoptosis in senescent cells;Reduction of the senescence-associated secretory phenotype (SASP);Promotion of tissue regeneration by creating space for healthy cells to proliferate.

Anyway, the real challenges are the trials of using these results obtained from animal experiments in clinical studies on humans in such a way that the results are maximized and the adverse effects are meaningless [112].

Geroprotectors are compounds that aim to slow the aging process by targeting multiple hallmarks of aging at the same time (Table 1). These include the following:Nutrient-sensing pathways;Cellular senescence;Mitochondrial dysfunction;Genomic instability;Epigenetic alterations.

For instance, Blagosklonny (2019) proposed that rapamycin’s geroprotective effects stem from its ability to suppress the hypertrophic phenotype associated with cellular senescence and thereby prevent the development of the SASP [122]. Recent research has also focused on the potential synergistic effects of combining senolytics and geroprotectors. For example, Xu et al. (2015) demonstrated that combining the senolytic dasatinib with the geroprotector rapamycin led to superior outcomes in alleviating age-related pathologies compared to either intervention alone [123]. This combination approach targets both the elimination of existing senescent cells and the prevention of new senescent cell accumulation, potentially offering a more comprehensive strategy for combating fibroblast senescence.

## 6. Discussion and Conclusions

The most important factor is to find strategies to combat the negative effects of fibroblast senescence [124]. Developing effective strategies to combat the negative effects of fibroblast senescence is crucial.

The current research focuses on exploring circulating factors associated with the senescence-associated secretory phenotype (SASP), such as interleukin-6 (IL-6) and matrix metalloproteinase-3 (MMP-3), as well as epigenetic markers like DNA methylation patterns, as exemplified by Horvath’s epigenetic clock and diagnostics [125]. Given the diverse nature of senescent cells across various tissue types, the development of tissue-specific senolytics and geroprotectors is mandatory. This approach could potentially improve therapeutic efficacy of these methods while reducing unintended side effects. This may involve a more detailed exploitation of senescence mechanisms for each tissue type, concomitantly with targeted drug administration, considering, of course, the optimal timing and doses for both senolytic drugs and geroprotectors. Intermittent dosing of senolytics has shown promise in pre-clinical studies; however, comprehensive long-term safety profiles and efficacy data in human subjects remain to be established [126,127].

Another research direction involves exploring the combination of senolytics and geroprotectors alongside the use of other anti-aging strategies such as physical exercises and dietary regimens, with the hope of developing more effective anti-aging strategies [128]. A significant challenge in the field remains the translation of preclinical findings to human clinical trials. Issues such as inter-species differences in aging mechanisms and the long-term nature of aging studies pose challenges for translational research. As research progresses towards potential lifespan extension in humans, ethical debates surrounding access, equity, and the societal implications of such interventions will become more and more important [129].

In conclusion, the use of fibroblasts and senolytics to combat the negative effects of senescence represents a promising direction in aging research. Their therapeutic use offers a longer lifespan and also extends patients healthspan, as they target the fundamental mechanisms of cellular aging. As our understanding of the molecular mechanisms underlying senescence and aging becomes complete, the effectiveness of treatments will increase, and the results will be more conclusive. The integration of advanced technologies, such as artificial intelligence, can accelerate progress in this field. Ultimately, the successful development and implementation of these strategies could have profound implications for human health and longevity, and would potentially revolutionize our approach to age-related diseases and the aging process itself [130].

Regarding skin aging, the two main contributing factors are the accumulation of senescent fibroblasts and the subsequent changes in the extracellular matrix. Senescent fibroblasts exhibit a distinct senescence-associated secretory phenotype (SASP), characterized by the release of pro-inflammatory cytokines, matrix metalloproteinases, and other bioactive molecules. This SASP further exacerbates ECM degradation, impairs wound healing, and promotes chronic inflammation—all of which negatively impact skin homeostasis and function [131].

The phenomenon of decreased extracellular matrix synthesis associated with aging, combined with its increased degradation, leads to changes in certain properties of the skin, such as reduced elasticity, accentuated wrinkles, dryness, and impaired capacity to heal dermal wounds [132]. Diminished collagen and elastin production, along with increased fragmentation and cross-linking, compromise the structural integrity and biomechanical properties of the dermal ECM. Additionally, reduced glycosaminoglycan synthesis decreases skin hydration, further contributing to an aged appearance [133].

To prevent and combat the negative effects of dermal fibroblast senescence, there are two truly promising therapeutic options: senolytics and geroprotectors. Senolytics are compounds that selectively induce apoptosis in senescent cells, thereby reducing the burden of the SASP and creating space for the proliferation of healthy cells [134].

Geroprotectors, on the other hand, aim to slow the aging process by targeting multiple hallmarks of aging, such as nutrient-sensing pathways, mitochondrial dysfunction, and epigenetic alterations [125].

By combining these two classes of substances—senolytics and geroprotectors—superior results have been demonstrated in alleviating age-related conditions, compared to the outcomes achieved by using only one type of substance.

One strategy focuses on eliminating existing senescent cells and preventing the accumulation of new ones. Studies investigating this strategy aim to combat fibroblast senescence and, consequently, reduce its impact on the skin [135].

However, significant challenges remain in translating these findings to human therapies. Nowadays, several test methods have been developed to detect biomarkers [136].

Developing reliable biomarkers and diagnostic tools for cellular senescence is crucial for assessing the efficacy of these interventions. In the case of most markers, care is recommended in interpreting the results, due to the individual variability given by diet, exercise, and the time of day at which the biological sample was collected. Additionally, determining the optimal timing and dosing regimens, as well as addressing potential off-target effects, is essential for the successful clinical implementation of senolytics and geroprotectors. Future studies should focus on translating the promising preclinical findings regarding senolytics and geroprotectors into effective and safe human therapies for skin aging, while paying particular attention to personalized approaches that account for individual variations in aging mechanisms and responses. Further research is needed to elucidate the intricate interplay between dermal fibroblast plasticity, cellular senescence, and ECM remodeling in the context of skin aging, particularly in identifying novel molecular targets and biomarkers that can predict and monitor the effectiveness of interventions.

## Figures and Tables

**Figure 1 biomedicines-13-01927-f001:**
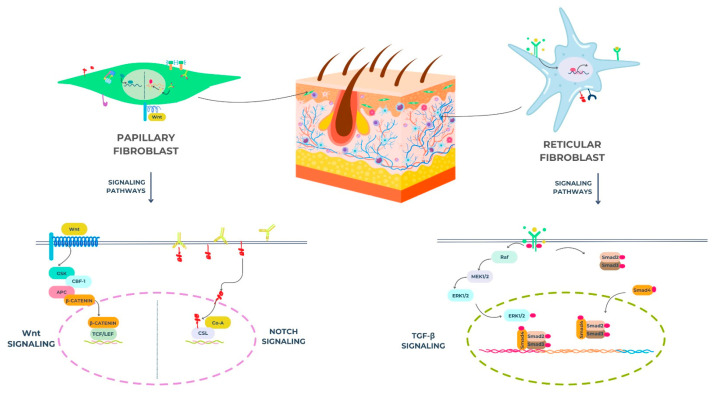
Signaling pathways in skin fibroblasts—function and interaction (figure made by the authors).

**Figure 2 biomedicines-13-01927-f002:**
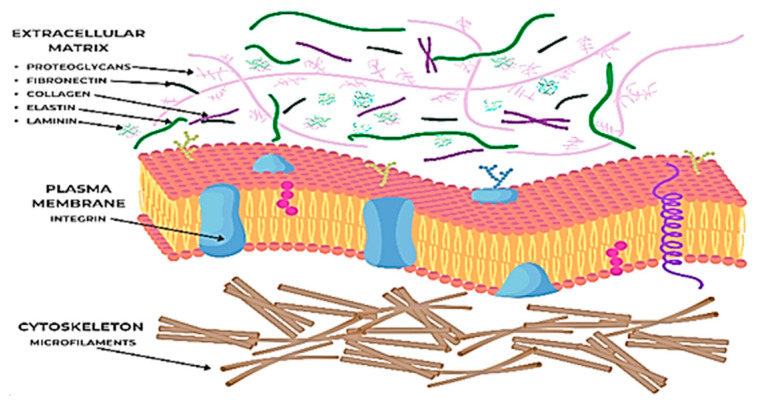
Components of extracellular matrix (ECM) (figure made by the authors).

**Figure 3 biomedicines-13-01927-f003:**
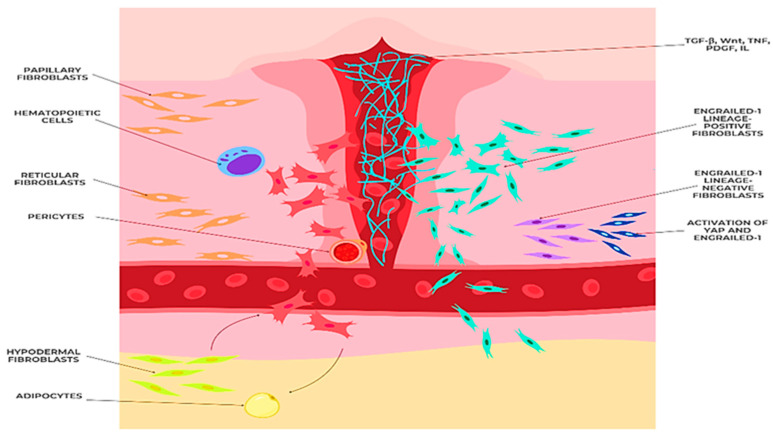
Mechanisms of fibroblast response in tissue regeneration (figure made by the authors).

**Figure 4 biomedicines-13-01927-f004:**
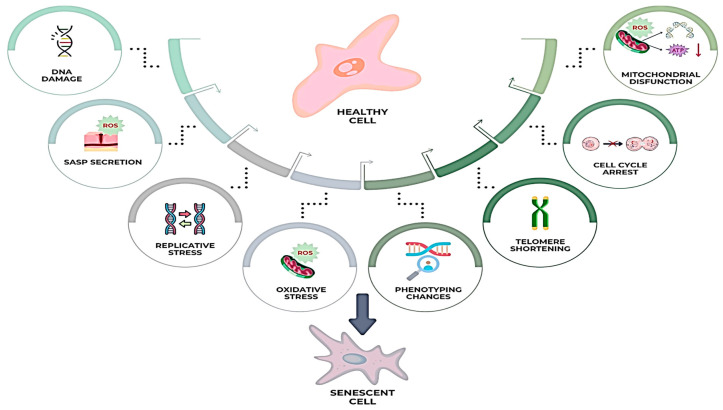
Phenotypic changes during dermal fibroblasts’ senescence.

**Table 1 biomedicines-13-01927-t001:** Senolytics and geroprotectors: examples, mechanisms of action, and target cell type.

Category	Medications	Mechanism of Action	Target Cell Types	Efficacy	References
**Senolytics**	Dasatinib, Quercetin	Inducing apoptosis in senescent cells	Senescent cells from various tissues	Significant reduction of sarcopenia	[113]
	Navitoclax	Blocking BCL-2 signaling to promote apoptosis	Senescent cells, especially from adipose tissue	Improvement of muscle function	[114,115]
**Geroprotectors**	Rapamycin, Metformin	Inhibiting the mTOR pathway to improve longevity	Cells from different types of tissue	Delaying the aging process	[116,117]
	Curcumin	Activating the Nrf2 pathway to reduce oxidative stress	Cells in general, especially those involved in inflammation	Reducing inflammation and oxidative stress	[118,119,120]
	Resveratrol	Activating the SIRT1 protein to induce longevity	Cells from all tissues, including muscular and nervous	Improving metabolism and cardiovascular function	[121,122]

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
