# Peer review of "Targeting Dermal Fibroblast Senescence: From Cellular Plasticity to Anti-Aging Therapies"

_biomedicines, 2025, doi:10.3390/biomedicines13081927_

Round 1
Reviewer 1 Report
Comments and Suggestions for Authors
Dermal fibroblast plasticity is maintained by intricate regulatory systems that allow for rapid response to injury and environmental challenges. A complex control system is responsible for determining skin health, aging, disease progression, and is based on skin fibroblast heterogeneity, metabolism, epigenetic regulation, and mechanotransduction. The understanding of the mechanisms that control this plasticity is essential for the development of targeted therapies for skin aging, wound healing disorders, and fibrotic diseases where dermal fibroblast dysfunction is a central factor.
- The title does not reflect the content of this paper. The authors indicate that “this review focuses on the role of dermal fibroblasts in skin aging and the possible molecular mechanisms involved in this process” [lines 55-56]. However, the title does not contain any mention of skin aging…
- The review's goals, objectives, and intended readers are not specified.
- Dermal fibroblasts differ from those in other tissues in their extraordinary functional diversity. Different subpopulations of dermal fibroblasts have unique molecular signatures and specialized functions. The paper lacks details about dermal fibroblasts and their subpopulations.
- There isn't enough modern information about the key signaling pathways that control fibroblast behavior and plasticity. Section 2.1 Key signaling pathways contains old references prior to 2021, some of them older than 2015 and not related to skin fibroblasts [ref.6,9].
- The list of literature consists of quite old works, and relatively new works do not reflect the stated topic (68% in the last 10 years and only 32% in the last 5 years).
- References do not conform to the stated facts about skin and contain general information or are not relative to dermal fibroblasts (for example ref. 6 is devoted to liver and gastrointestinal cancer; ref. 9, 16, 17 about lung fibrosis, ref. 11 about dental medicine, ref. 21 about cartilage, ref. 27 about urethral scar formation etc.). Ref. 69 does not have complete information about the journal, volume, pages, etc.
- The tables do not contain any reference to the source of stated information in cells.
In my view, the article's structure should be updated to incorporate modern data on the characteristics of skin fibroblasts and their subpopulations, mechanisms that govern their plasticity and aging, and updating the literature list.
Comments on the Quality of English LanguageThe work has numerous grammatical and stylistic errors that require correction
Author Response
Dear reviewer,
Thank you for your time and consideration in evaluating our manuscript, entitled " Plasticity of Dermal Fibroblasts: A Multifaceted Approach ". We particularly appreciate the constructive comments and suggestions provided, which have helped us to significantly improve the quality of our work.
Below you will find our point-by-point responses to the recommendations you provided, highlighted in yellow within the article text.
Comments 1: The title does not reflect the content of this paper. The authors indicate that “this review focuses on the role of dermal fibroblasts in skin aging and the possible molecular mechanisms involved in this process” [lines 55-56]. However, the title does not contain any mention of skin aging…
Response 1: We appreciate your suggestion regarding the title. We agree that the original title did not adequately reflect the focus on skin aging, and we have revised it to ‘’ Targeting Dermal Fibroblast Senescence: From Cellular Plasticity to Anti-Aging Therapies’’ for better reflect to the scope and content of our manuscript.
Comments 2: The review's goals, objectives, and intended readers are not specified.
Response 1: Thank you very much for pointing this out. To clarify the aims and scope of our review, we have added explicit statements in the Introduction section outlining the goals and objectives of the manuscript. This information has been added to the final paragraph of the introduction.
This review aims to provide a comprehensive and up-to-date overview of dermal fibroblast plasticity and senescence and their impact on skin aging, with a secondary goal of highlighting potential therapeutic targets for combating age-related skin changes. To achieve this, we examine the molecular mechanisms underlying dermal fibroblast plasticity, discuss the impact of aging on ECM synthesis and remodeling, explore the role of cellular senescence in dermal fibroblasts, and present therapeutic perspectives focusing on senolytics and geroprotectors. This review is intended for researchers in the fields of dermatology, cell biology, aging research, and regenerative medicine seeking an overview of the topic.
Comments 3: Dermal fibroblasts differ from those in other tissues in their extraordinary functional diversity. Different subpopulations of dermal fibroblasts have unique molecular signatures and specialized functions. The paper lacks details about dermal fibroblasts and their subpopulations.
Response 3: We thank you for this valuable comment. We agree that a more detailed discussion of dermal fibroblast subpopulations is important for understanding their functional diversity and roles in skin aging. In response, we have expanded the relevant section to include additional information about the distinct subtypes of dermal fibroblasts. Specifically:
- We have added a statement in the Introduction emphasizing that dermal fibroblasts represent a heterogeneous population composed of distinct subtypes with specialized roles.
- In Section 2, we introduced a new subsection (4: Signaling Pathways and Dermal Fibroblast Subpopulations) to discuss how different subpopulations—such as papillary and reticular fibroblasts—may exhibit differential responses to key signaling pathways (e.g., Wnt, TGF-β, PDGF).
- In Section 3, we included a new subsection (3: ECM Composition and Dermal Fibroblast Subpopulations) that explores how these subpopulations differ in their ECM production, molecular markers, spatial localization, and contributions to tissue structure and repair.
1.Introduction
It is important to note that dermal fibroblasts are not a homogeneous population, but consist of distinct subpopulations with unique molecular signatures and specialized functions. For example, fibroblasts in the papillary dermis, characterized by the expression of Dkk3 and CD90, differ from those in the reticular dermis, which express higher levels of elastin and fibulin-1, in their expression of ECM components, response to growth factors, and role in wound healing [6]. Furthermore, some fibroblast populations are localized to specific skin layers depending on their cellular origin, such as pre-adipocytes giving rise to fibroblasts after injury [5]. A comprehensive understanding of dermal fibroblast plasticity requires consideration of these specialized subpopulations and their unique characteristics.
New Subsection in Section 2: Signaling Pathways and Dermal Fibroblast Subpopulations:
2.4 Signaling Pathways and Dermal Fibroblast Subpopulations
While the signaling pathways described above (TGF-β, PDGF, Wnt, FGF) are relevant to all dermal fibroblasts, it's increasingly evident that different subpopulations may exhibit distinct responses to these signals. For instance, papillary fibroblasts, being more responsive to Wnt signaling, have been shown to exhibit increased proliferation and migration in response to Wnt ligands compared to reticular fibroblasts [33].Reticular fibroblasts, in contrast, may be more sensitive to TGF-β, leading to increased collagen production and a greater propensity for scar formation [34]. Furthermore, the expression of specific signaling receptors, such as the PDGF receptor alpha (PDGFRA), may vary among fibroblast subpopulations, leading to differences in their responses to PDGF [35]. Further research is needed to fully elucidate the specific signaling mechanisms that regulate the unique functions of different dermal fibroblast subpopulations.
New Subsection in Section 3: ECM Composition and Dermal Fibroblast Subpopulations:
3.3 ECM Composition and Dermal Fibroblast Subpopulations
The ECM is not only a product of fibroblast activity, but also a key regulator of fibroblast phenotype and function. Distinct dermal fibroblast subpopulations contribute differently to the composition and organization of the ECM. Papillary fibroblasts, characterized by their stellate morphology and proximity to the epidermis, are primarily responsible for synthesizing type III collagen and components of the basement membrane, such as laminin and collagen IV, contributing to the structural integrity of the epidermal-dermal junction and providing support for keratinocytes [37]. Reticular fibroblasts, which are larger and more elongated, are the main producers of type I collagen, providing tensile strength to the deeper layers of the dermis and organizing the collagen fibers into thick bundles [38].Furthermore, the specific ECM microenvironment surrounding different fibroblast subpopulations, such as the presence of hyaluronic acid in the papillary dermis, can influence their response to growth factors and cytokines, creating a feedback loop that further shapes their phenotype and function. In addition, following injury, pre-adipocytes transdifferentiate to fibroblasts and deposit ECM at the site [39].
Comments 4: There isn't enough modern information about the key signaling pathways that control fibroblast behavior and plasticity. Section 2.1 Key signaling pathways contains old references prior to 2021, some of them older than 2015 and not related to skin fibroblasts [ref.6,9].
Response 4: We appreciate the reviewer’s observation regarding the outdated references in Section 2.1. In response, we have revised this section to incorporate more recent and relevant studies, particularly those addressing the specific roles of key signaling pathways in dermal fibroblasts. References 6 and 9, which were identified as outdated and less relevant, have been replaced as follows:
- Reference [7]: Kang et al., FASEB Journal (2020) — describing TGF-β–induced PD-L1 expression and release via extracellular vesicles.
- Reference [11]:Sun et al., Experimental and Therapeutic Medicine (2024) — detailing PDGF signaling in the proliferation and migration of human adipose-derived stem cells.
Additionally, Section 2.1 has been substantially revised to include updated mechanistic insights and more recent bibliographic references, thereby strengthening the scientific relevance and accuracy of the manuscript.
These key signaling pathways play essential roles in regulating fibroblast behavior and plasticity, contributing to tissue homeostasis and repair in normal conditions. However, dysregulation of these pathways can lead to a variety of pathological conditions, highlighting the importance of understanding their dual roles in both health and disease. These pathways are often interconnected and influence each other, creating a complex regulatory network.
- Transforming Growth Factor-beta (TGF-β) Signaling:
TGF-β is a pleiotropic cytokine with diverse effects on fibroblasts. In physiological conditions, TGF-β plays a key role in regulating extracellular matrix (ECM) production, cell proliferation, and differentiation [7], ensuring proper tissue structure and repair. TGF-β signaling activates SMAD proteins, leading to transcriptional regulation of genes involved in ECM synthesis (e.g., collagen, fibronectin) and other fibroblast functions.
However, dysregulation of TGF-β signaling has been linked to fibrosis and other pathological conditions [8]. Excessive activation of TGF-β can lead to overproduction of ECM components, resulting in tissue fibrosis in various organs, including the skin. This dysregulation can be triggered by chronic inflammation, genetic factors, or other environmental stimuli [9].
- Fibroblast Growth Factor (FGF) Signaling:
FGFs are a family of growth factors that bind to FGF receptors and activate downstream signaling cascades involving MAPK/ERK and PI3K/Akt pathways[8]. Under normal conditions, FGF signaling pathways influence cell proliferation, migration, and differentiation, which is essential for wound healing and tissue repair.
However, in pathological conditions, sustained FGF signaling can contribute to tumor growth and angiogenesis [11].In the context of fibrosis, excessive FGF signaling can promote fibroblast proliferation and ECM deposition, contributing to tissue scarring.
- Platelet-Derived Growth Factor (PDGF) Signaling:
PDGF is a potent mitogen for fibroblasts, stimulating cell proliferation and migration [12]. PDGF signaling involves the activation of receptor tyrosine kinases and downstream signaling through PI3K/Akt and MAPK/ERK pathways [13]. Physiologically, PDGF plays a critical role in wound healing and tissue regeneration [14], promoting fibroblast recruitment and proliferation at the site of injury.
In pathological conditions, sustained or excessive PDGF signaling can contribute to conditions such as hypertrophic scarring and fibrosis. Overexpression of PDGF can lead to excessive fibroblast proliferation and ECM deposition, disrupting normal tissue architecture [15].
- Wnt Signaling:
The Wnt signaling pathway is involved in regulating various aspects of fibroblast behavior, including proliferation, differentiation, and ECM production [16]. Disruptions in Wnt signaling have been implicated in various skin diseases and aging processes. In healthy skin, Wnt signaling helps maintain proper cell differentiation, tissue architecture, and ECM production.
However, abnormal Wnt signaling can lead to dysregulated fibroblast functions, contributing to skin conditions such as fibrosis, psoriasis, and atopic dermatitis by affecting collagen deposition and ECM remodeling in aging. For example, increased Wnt signaling can promote excessive collagen production and fibrosis, while decreased signaling can impair wound healing and tissue regeneration. As organisms age, Wnt signaling can become dysregulated, impacting fibroblast functionality and leading to skin aging characteristics, such as reduced elasticity, increased sagging, and a diminished capacity for wound healing.
In conclusion, these signaling pathways exhibit remarkable versatility, playing crucial roles in both maintaining tissue homeostasis and driving pathological processes when dysregulated. A comprehensive understanding of these dual functions is essential for developing targeted therapeutic strategies that can selectively modulate pathway activity in disease while preserving their normal physiological roles.
Comments 5: The list of literature consists of quite old works, and relatively new works do not reflect the stated topic (68% in the last 10 years and only 32% in the last 5 years).
Response 5: We appreciate your feedback regarding our reference list. We have thoroughly reviewed the citations, replacing older references with more recent and relevant publications whenever possible, and ensuring that all included references directly support the claims made in our manuscript. The percentage of articles older than or from 2020 in the bibliography is approximately 23%.
Comments 6: References do not conform to the stated facts about skin and contain general information or are not relative to dermal fibroblasts (for example ref. 6 is devoted to liver and gastrointestinal cancer; ref. 9, 16, 17 about lung fibrosis, ref. 11 about dental medicine, ref. 21 about cartilage, ref. 27 about urethral scar formation etc.). Ref. 69 does not have complete information about the journal, volume, pages, etc.
Response 6: We appreciate your thorough review of our reference list. We have carefully examined all citations and removed those not directly relevant to dermal fibroblast plasticity and skin aging. Where possible, we replaced older references with more recent and pertinent publications. However, we retained some older references that provide essential historical context or describe foundational mechanisms still relevant today. All key points are supported by up-to-date literature to ensure accuracy and relevance. Additionally, we have corrected incomplete citation details, including for reference 69.
We replaced reference nr 6
Katz, L.H.; Likhter, M.; Jogunoori, W.; Belkin, M.; Ohshiro, K.; Mishra, L. TGF-β Signaling in Liver and Gastrointestinal Cancers. Cancer Lett 2016, 379, 166–172, doi:10.1016/j.canlet.2016.03.033.
With reference nr 7
Kang, J.-H.; Jung, M.-Y.; Choudhury, M.; Leof, E.B. Transforming Growth Factor Beta Induces Fibroblasts to Express and Release the Immunomodulatory Protein PD-L1 into Extracellular Vesicles. FASEB J 2020, 34, 2213–2226, doi:10.1096/fj.201902354R.
We replaced reference nr 9
Noskovičová, N.; Petřek, M.; Eickelberg, O.; Heinzelmann, K. Platelet-Derived Growth Factor Signaling in the Lung. From Lung Development and Disease to Clinical Studies. Am J Respir Cell Mol Biol 2015, 52, 263–284, doi:10.1165/rcmb.2014-0294TR.
With reference nr 11
Sun, Z.; Fukui, M.; Taketani, S.; Kako, A.; Kunieda, S.; Kakudo, N. Predominant Control of PDGF/PDGF Receptor Signaling in the Migration and Proliferation of Human Adipose‑derived Stem Cells under Culture Conditions with a Combination of Growth Factors. Experimental and Therapeutic Medicine 2024, 27, 1–14, doi:10.3892/etm.2024.12444.
About the reference nr 16 and 17 we mention the general concept of histone acetylation and its impact on gene expression. These references provide a comprehensive review of histone acetylation mechanisms, though with a focus on their role in cancer. We included this reference because it offers a valuable overview of the fundamental principles of histone acetylation, which are relevant to our discussion of epigenetic regulation in fibroblasts.
We thank you for identifying that reference 11 was not directly relevant to the text. It has been removed from the manuscript.
We included reference nr [29] in the proteoglycans section as it provides a comprehensive introduction to the structure, biosynthesis, and degradation of these macromolecules. Although the article's primary focus is on cartilage proteoglycans, the fundamental principles of proteoglycan metabolism described within are broadly applicable to proteoglycans in other tissues, including the dermis. This reference offers valuable background information for understanding how fibroblasts regulate ECM turnover through the synthesis and breakdown of proteoglycans.
We have revised the manuscript to replace reference [27] Sa, Y.; Li, C.; Li, H.; Guo, H. TIMP-1 Induces α-Smooth Muscle Actin in Fibroblasts to Promote Urethral Scar Formation. Cell Physiol Biochem 2015, 35, 2233–2243, doi:10.1159/000374028.
with [35] Brew, K.; Nagase, H. The Tissue Inhibitors of Metalloproteinases (TIMPs): An Ancient Family with Structural and Functional Diversity. Biochimica et Biophysica Acta (BBA) - Molecular Cell Research 2010, 1803, 55–71, doi:10.1016/j.bbamcr.2010.01.003.
We acknowledge that the original reference contained primarily information on urethral scar tissue and the specific mechanisms involved and of course that context make it less generally applicable to skin aging., whereas the new reference provides more specific and relevant data directly related to specific aspect of dermal fibroblasts.
We decided to also remove ref 69.
Comments 7: The tables do not contain any reference to the source of stated information in cells.
Response 7: Thank you for your valuable comment regarding the tables. We have revised the tables to include appropriate references for the information presented in each cell, ensuring that all data are properly sourced and supported by the relevant literature.
|
Category |
Medications |
Mechanism of Action |
Target Cell Types |
Efficacy |
References |
|
Senolytics |
Dasatinib, Quercetin |
Inducing apoptosis in senescent cells |
Senescent cells from various tissues |
Significant reduction of sarcopenia |
[113] |
|
|
Navitoclax |
Blocking BCL-2 signaling to promote apoptosis |
Senescent cells, especially from adipose tissue |
Improvement of muscle function |
[114,115] |
|
Geroprotectors |
Rapamycin, Metformin |
Inhibiting the mTOR pathway to improve longevity |
Cells from different types of tissue |
Delaying the aging process |
[116,117] |
|
|
Curcumin |
Activating the Nrf2 pathway to reduce oxidative stress |
Cells in general, especially those involved in inflammation |
Reducing inflammation and oxidative stress |
[118-120] |
|
|
Resveratrol |
Activating the SIRT1 protein to induce longevity |
Cells from all tissues, including muscular and nervous |
Improving metabolism and cardiovascular function |
[121,122] |
- Orioli D, Dellambra E. Epigenetic Regulation of Skin Cells in Natural Aging and Premature Aging Diseases. Cells. 2018 Dec 12;7(12):268. doi: 10.3390/cells7120268. PMID: 30545089; PMCID: PMC6315602.
- Zheng L, He S, Wang H, Li J, Liu Y, Liu S. Targeting Cellular Senescence in Aging and Age-Related Diseases: Challenges, Considerations, and the Emerging Role of Senolytic and Senomorphic Therapies. Aging Dis. 2024 Feb 27;15(6):2554-2594. doi: 10.14336/AD.2024.0206. PMID: 38421832; PMCID: PMC11567261.
- Zhu Y, Tchkonia T, Fuhrmann-Stroissnigg H, Dai HM, Ling YY, Stout MB, Pirtskhalava T, Giorgadze N, Johnson KO, Giles CB, Wren JD, Niedernhofer LJ, Robbins PD, Kirkland JL. Identification of a novel senolytic agent, navitoclax, targeting the Bcl-2 family of anti-apoptotic factors. Aging Cell. 2016 Jun;15(3):428-35. doi: 10.1111/acel.12445. Epub 2016 Mar 18. PMID: 26711051; PMCID: PMC4854923..
- Elliehausen CJ, Anderson RM, Diffee GM, Rhoads TW, Lamming DW, Hornberger TA, Konopka AR. Geroprotector drugs and exercise: friends or foes on healthy longevity? BMC Biol. 2023 Dec 8;21(1):287. doi: 10.1186/s12915-023-01779-9. PMID: 38066609; PMCID: PMC10709984.
- Le Couteur DG, Barzilai N. New horizons in life extension, healthspan extension and exceptional longevity. Age Ageing. 2022 Aug 2;51(8):afac156. doi: 10.1093/ageing/afac156. PMID: 35932241; PMCID: PMC9356533.
- Ashrafizadeh M, Ahmadi Z, Mohammadinejad R, Farkhondeh T, Samarghandian S. Curcumin Activates the Nrf2 Pathway and Induces Cellular Protection Against Oxidative Injury. Curr Mol Med. 2020;20(2):116-133. doi: 10.2174/1566524019666191016150757. PMID: 31622191.
- Cui J, Li H, Zhang T, Lin F, Chen M, Zhang G, Feng Z. Research progress on the mechanism of curcumin anti-oxidative stress based on signaling pathway. Front Pharmacol. 2025 Apr 7;16:1548073. doi: 10.3389/fphar.2025.1548073. PMID: 40260389; PMCID: PMC12009910.
- Liao, D., Shangguan, D., Wu, Y. et al. Curcumin protects against doxorubicin induced oxidative stress by regulating the Keap1-Nrf2-ARE and autophagy signaling pathways. Psychopharmacology 240, 1179–1190 (2023). https://doi.org/10.1007/s00213-023-06357-z
- Zhang W, Qian S, Tang B, Kang P, Zhang H, Shi C. Resveratrol inhibits ferroptosis and decelerates heart failure progression via Sirt1/p53 pathway activation. J Cell Mol Med. 2023 Oct;27(20):3075-3089. doi: 10.1111/jcmm.17874. Epub 2023 Jul 24. PMID: 37487007; PMCID: PMC10568670.
- Blagosklonny, M.V. Rapamycin for Longevity: Opinion Article. Aging (Albany NY) 2019, 11, 8048–8067, doi:10.18632/aging.102355.

Reviewer 2 Report
Comments and Suggestions for Authors
This manuscript offers a comprehensive review of the plasticity of dermal fibroblasts and their role in skin aging, with particular emphasis on ECM remodeling and cellular senescence. The authors highlight the molecular mechanisms, including key signalling pathways, epigenetic modifications, and microRNA involvement. The topic is relevant, but major revision is necessary to enhance its clarity and presentation.
Introduction
Several ideas are revisited redundantly, for example fibroblast role in ECM.
Section 2.
In the section “Molecular mechanisms” distinguish between physiological and pathological roles of TGF-β, PDGF, Wnt, FGF.
Section 3.
Add specific examples related to microRNA section.
Clarify the regulatory balance between MMPs and TIMPs.
The other sections are well structured.
Language:
Multiple grammar and syntax issues are present
In addition, the entire manuscript requests a formatting according to the guidelines. The authors should add some figures more explicative and not simple schemes.
Author Response
Dear reviewer,
Thank you to you and the reviewers for your time and consideration in evaluating our manuscript, iniatialy entitled " Plasticity of Dermal Fibroblasts: A Multifaceted Approach ". We particularly appreciate the constructive comments and suggestions provided, which have helped us to significantly improve the quality of our work. At the suggestion of another reviewer who analyzed the manuscript, we decided to change the title in ‘’Targeting Dermal Fibroblast Senescence: From Cellular Plasticity to Anti-Aging Therapies’’ for better reflect to the scope and content of our manuscript.
Below, we address each of the requests and observations raised by the you point by point.
Comments 1.Introduction-several ideas are revisited redundantly, for example fibroblast role in ECM
Response 1:In accordance with your suggestion, we have highlighted the key findings in the abstract.
We have carefully revised the section to eliminate redundancy and improve the flow of information. We removed the sentence:’’ Cellular plasticity generally refers to the ability of cells to adapt their phenotype and function based on their microenvironment’’ as well as ‘’This phenotypic flexibility is essential for fibroblasts to adapt to the needs of the tissue and enables them to contribute to diverse biological processes.’’
In addition,we replaced the following paragraph:This review focuses on the role of dermal fibroblasts in skin aging and and the possible molecular mechanisms involved in this process. Dermal fibroblasts, mesodermal cells essential for maintaining the integrity of connective tissue, exhibit remarkable plasticity, adapting their phenotype and function to diverse conditions. We will analyze the plasticity of dermal fibroblasts, the impact of senescence on their functions, and their contribution to tissue homeostasis and repair in the context of aging. Dermal fibroblasts, the predominant stromal cells of the dermis, play an essential role in maintaining tissue integrity and homeostasis. Unlike epithelial cells, which are polarized and form continuous layers, fibroblasts are mesenchymal cells characterized by remarkable plasticity ….
with the revised version : Dermal fibroblasts, the predominant stromal cells of the dermis, exhibit remarkable plasticity, adapting their phenotype and function in response to diverse stimuli. This plasticity allows them to play essential roles in tissue homeostasis, wound healing, and extracellular matrix (ECM) production and remodeling. Unlike epithelial cells, which are polarized and form continuous layers, fibroblasts are mesenchymal cells characterized by this remarkable adaptability, enabling them to contribute to a wide range of biological processes .
We also added the following sentence to the final version of the Introduction to enhance conceptual clarity: This phenotypic flexibility is crucial for maintaining dermal integrity, enabling fibroblasts to respond effectively to the changing needs of the tissue during aging, injury, or disease .
At the end of the introduction we added two new paragraphs section outlining the goals and objectives of the manuscript.
It is important to note that dermal fibroblasts are not a homogeneous population, but consist of distinct subpopulations with unique molecular signatures and specialized functions. For example, fibroblasts in the papillary dermis, characterized by the expression of Dkk3 and CD90, differ from those in the reticular dermis, which express higher levels of elastin and fibulin-1, in their expression of ECM components, response to growth factors, and role in wound healing [6]. Furthermore, some fibroblast populations are localized to specific skin layers depending on their cellular origin, such as pre-adipocytes giving rise to fibroblasts after injury [5]. A comprehensive understanding of dermal fibroblast plasticity requires consideration of these specialized subpopulations and their unique characteristics.
Lastly, we clarified the aims of the review in the final paragraph of the Introduction:
This review aims to provide a comprehensive and up-to-date overview of dermal fibroblast plasticity and senescence and their impact on skin aging, with a secondary goal of highlighting potential therapeutic targets for combating age-related skin changes. To achieve this, we examine the molecular mechanisms underlying dermal fibroblast plasticity, discuss the impact of aging on ECM synthesis and remodeling, explore the role of cellular senescence in dermal fibroblasts, and present therapeutic perspectives focusing on senolytics and geroprotectors. This review is intended for researchers in the fields of dermatology, cell biology, aging research, and regenerative medicine seeking an overview of the topic.
Comments 2. In the section “Molecular mechanisms” distinguish between physiological and pathological roles of TGF-β, PDGF, Wnt, FGF.
Response 2: Thank you for pointing this out. We revised Section 2 to better differentiate between the physiological and pathological roles of TGF-β, PDGF, Wnt, and FGF.theese are the new paragraphs that we introduced.
In physiological conditions, TGF-β plays a key role in regulating extracellular matrix (ECM) production, cell proliferation, and differentiation [7], ensuring proper tissue structure and repair. TGF-β signaling activates SMAD proteins, leading to transcriptional regulation of genes involved in ECM synthesis (e.g., collagen, fibronectin) and other fibroblast functions [8].However, dysregulation of TGF-β signaling has been linked to fibrosis and other pathological conditions . Excessive activation of TGF-β can lead to overproduction of ECM components, resulting in tissue fibrosis in various organs, including the skin. This dysregulation can be triggered by chronic inflammation, genetic factors, or other environmental stimuli [9].
Under normal conditions, FGF signaling pathways influence cell proliferation, migration, and differentiation, which is essential for wound healing and tissue repair.
However, in pathological conditions, sustained FGF signaling can contribute to tumor growth and angiogenesis [11]. In the context of fibrosis, excessive FGF signaling can promote fibroblast proliferation and ECM deposition, contributing to tissue scarring
Physiologically, PDGF plays a critical role in wound healing and tissue regeneration promoting fibroblast recruitment and proliferation at the site of injury. In pathological conditions, sustained or excessive PDGF signaling can contribute to conditions such as hypertrophic scarring and fibrosis. Overexpression of PDGF can lead to excessive fibroblast proliferation and ECM deposition, disrupting normal tissue architecture [15].
In healthy skin, Wnt signaling helps maintain proper cell differentiation, tissue architecture, and ECM production.Abnormal Wnt signaling can lead to dysregulated fibroblast functions, contributing to skin conditions such as fibrosis, psoriasis, and atopic dermatitis by affecting collagen deposition and ECM remodeling in aging. For example, increased Wnt signaling can promote excessive collagen production and fibrosis, while decreased signaling can impair wound healing and tissue regeneration.
In conclusion, these signaling pathways exhibit remarkable versatility, playing crucial roles in both maintaining tissue homeostasis and driving pathological processes when dysregulated. A comprehensive understanding of these dual functions is essential for developing targeted therapeutic strategies that can selectively modulate pathway activity in disease while preserving their normal physiological roles.
Comments 3.Add specific examples related to microRNA section.Clarify the regulatory balance between MMPs and TIMPs.
Response 3: We appreciate your feedback on Section 2.3. We have revised the text to address your suggestions and included specific examples related to microRNAs.
For example, the miR-29 family (including miR-29a, miR-29b, and miR-29c) has been shown to play a critical role in regulating collagen expression. These miRNAs directly target mRNAs encoding collagen types I and III, and their downregulation in fibrotic conditions leads to increased collagen synthesis and ECM deposition [2. Conversely, miR-21 is often upregulated in fibrotic environments, promoting fibroblast proliferation and migration by targeting genes involved in apoptosis and cell cycle regulation, such as PTEN and PDCD4 [25]. Furthermore, miR-155 has been implicated in the regulation of inflammation and ECM remodeling. Upregulation of miR-155 in fibroblasts can promote the expression of pro-inflammatory cytokines and MMPs, contributing to ECM degradation and tissue damage [26].Finally, miR-196a has been shown to regulate collagen expression and fibrosis by targeting genes involved in TGF-β signaling pathway[27]. These are just a few examples of the many miRNAs that contribute to fibroblast plasticity, highlighting the complex and multifaceted nature of this regulatory network.
Regarding the MMP/TIMP regulatory balance, we have expanded subsection 3.2 to clarify this dynamic equilibrium and its importance in maintaining ECM homeostasis. We also included additional mechanistic details regarding TIMP-mediated inhibition and the consequences of MMP/TIMP imbalance.
Fibroblasts are essential cells involved in building and remodeling the extracellular matrix (ECM), which is vital for keeping tissues structured and functioning properly. The ECM is in a dynamic balance between synthesis and degradation, and fibroblasts play a crucial role in maintaining this equilibrium. These cells are the first to produce ECM components and are responsible for constantly turning over and adjusting the matrix as needed, whether due to normal activity or in response to various physiological and pathological stimuli. Fibroblasts synthesize a large number of ECM proteins, including collagens, elastin, fibronectin, and proteoglycans, which come together to form the supportive framework [28]. In addition to manufacturing these structural proteins, fibroblasts also produce enzymes called matrix metalloproteinases (MMPs), which are responsible for breaking down components of the ECM. This degradation is crucial for ECM remodeling during tissue repair and normal turnover.MMPs are a family of zinc-dependent endopeptidases that degrade various components of the ECM. Their activity is carefully controlled by tissue inhibitors of metalloproteinases (TIMPs), which bind to MMPs and prevent excessive degradation. TIMPs inhibit MMP activity by binding to their active sites, preventing them from interacting with ECM substrates. This ensures that ECM degradation is tightly controlled, preventing excessive breakdown of the matrix. The balance between ECM creation and breakdown is tightly managed by fibroblasts, allowing tissues to stay healthy and adapt to different conditions [29]. This balance is dynamic, allowing for ECM remodeling during tissue development, wound healing, and normal tissue turnover. TIMPs are a family of proteins that bind to MMPs and inhibit their activity, maintaining ECM homeostasis [30].However, this balance can be disrupted in various pathological conditions. For example, in skin aging, increased MMP activity and decreased TIMP expression contribute to the degradation of collagen and elastin fibers, leading to wrinkle formation and loss of skin elasticity. The balance between ECM synthesis and degradation is tightly controlled by fibroblasts, allowing for the maintenance of tissue homeostasis and adaptation to changing environmental conditions. In response to injury or disease, fibroblasts can become activated, leading to increased ECM production and remodeling. This process is critical for wound healing and tissue repair but can also contribute to fibrosis and scarring if dysregulated [31]. Fibroblasts also interact with other cell types, such as immune cells and endothelial cells, through the secretion of growth factors and cytokines, further influencing the ECM composition and tissue microenvironment. The ability of fibroblasts to sense and respond to mechanical forces through mechanotransduction pathways allows them to adapt ECM synthesis and remodeling to the mechanical needs of the tissue. Recent studies have shown that fibroblast populations are diverse within and between different tissues, suggesting that certain subsets may have specialized roles in regulating the ECM [32]. Understanding the complex relationship between fibroblasts and the ECM is essential for developing therapeutic strategies targeting fibrosis, tissue engineering, and regenerative medicine applications.Therefore, maintaining the appropriate MMP/TIMP regulatory balance is essential for ensuring proper ECM turnover, tissue homeostasis, and effective responses to injury or disease, making it a critical target for therapeutic interventions aimed at modulating tissue remodeling.
Comments 4: Multiple grammar and syntax issues are present
Response 4:Thank you for your comment.We have carefully revised the manuscript to correct grammar and syntax issues and improve overall language quality.
Comments 5: the authors should add some figures more explicative and not simple schemes.
Response 5: In addition to the textual revisions, we have also enhanced the manuscript with more explicit figures to visually represent the key concepts discussed. These figures will be included in the final version.
Figure nr 1: Signaling Pathways in Skin Fibroblast- Function and Interaction(figure made by the authors).
Figure 2. Extracellular Matrix (ECM): components and function (figure made by the authors).
Figure 3. Mechanisms of Fibroblast Response in Tissue Regeneration(figure made by the authors)
Figure nr 4:DNA Damage and Cell Senescence: Pathways of Aging (figure made by the authors)

Reviewer 3 Report
Comments and Suggestions for Authors
Title: can be more informative and more relevant to the topic
explain the main differences between your work and the following work: https://www.sciencedirect.com/science/article/pii/S156816372300154X
Abstract: it will be great if highlight some distinguished results in the abstract.
Section 2: needs images. try to show the mechanism by images
section 3-1: each paragraph is 1-2 lines. not suitable at all. Also this part is a general topic. it must be more specified.
Section 3: provide an interesting image.
many short paragraphs are used in the whole MS. They should be modified.
Section 4: also needs an image
sub-section of section 4 are too short. the authors should expand them by providing enough updated information from the previous studies.
Fig. 2 and Fig. 3 needs to be modified, suitable for a review paper.
Table 1, use number instead of Greek numbers. it needs citation if information extracted.
Also remove the blue color.
The MS needs a future perspective.
General comment:
This review paper looks like a report, and the context is not well organized. some sections are too short and not well explained.
Author Response
Dear reviewer,
We sincerely thank you for the thoughtful and constructive feedback on our manuscript, initially entitled "Plasticity of Dermal Fibroblasts: A Multifaceted Approach." Your comments have been truly helpful in refining and improving the overall quality of our work.
Below, you will find our point-by-point responses to each of your comments and suggestions.
Comments 1: Title: can be more informative and more relevant to the topic
Response 1: Thank you for your suggestion regarding the title. In response, we have revised it to: “Targeting Dermal Fibroblast Senescence: From Cellular Plasticity to Anti-Aging Therapies”, which we believe better reflects the focus and scope of the manuscript.
Comments 2: explain the main differences between your work and the following work: https://www.sciencedirect.com/science/article/pii/S156816372300154X
Response 2:In response to your request for finding differences between our manuscript and the related work by Salminen (2023) on fibroblast plasticity, we would like to clarify the following key differences:
Scope: Our manuscript is specifically focused on dermal fibroblasts and their role in skin aging, ECM remodeling, and senescence. We were interested into the molecular mechanisms and clinical consequences specific to this context.
Perspective: We emphasize the impact of aging on the synthesis and degradation of ECM components in the dermis, as well as the cellular and microenvironmental alterations involved. The Salminen review offers a broader perspective on fibroblast plasticity across various tissues and diseases beyond skin aging. It explores the origin, differentiation, and heterogeneity of fibroblasts in different pathological states, such as cancer and fibrosis.
While both manuscripts address the topic of fibroblast plasticity, our study is targeted towards the specific context of skin aging and dermal ECM remodeling, whereas the Salminen review offers a larger overview of the role of fibroblasts in various pathological conditions.
Comments 3: Abstract: it will be great if highlight some distinguished results in the abstract.
Response 3: In accordance with your suggestion, we have highlighted the key findings in the abstract using highlight text.
Abstract. In conclusion, understanding the complex interplay between dermal fibroblast plasticity, cellular senescence, and extracellular matrix (ECM) remodeling is essential for developing effective anti-aging interventions, highlighting the need for further research into senolytic and geroprotective therapies to enhance skin health and longevity. This approach has shown promising results in preclinical studies, demonstrating improved skin elasticity and reduced signs of aging .
Comments 4: Section 2: needs images. try to show the mechanism by images
Response 4: To address your comment, we have included two illustrative figure in Section 2, highlighting the key mechanisms and signaling pathways that regulate fibroblast behavior and plasticity.
Figure nr 1: Signaling Pathways in Skin Fibroblast- Function and Interaction . (figure made by the authors).
Comments 5: section 3-1: each paragraph is 1-2 lines. not suitable at all. Also this part is a general topic. it must be more specified.
Section 3: provide an interesting image.
Many short paragraphs are used in the whole MS. They should be modified.
Response 5: In Section 3, we have addressed your comments by expanding subsection 3.1 with additional information. We have also included a relevant figure to visually support the content."
In addition, we have revised the manuscript to merge several short paragraphs where appropriate, improving the overall flow and coherence of the text.
This complex network is not just a passive support; it plays an essential role in regulating cellular functions, including proliferation, differentiation, migration, cell shape, gene expression, and cell survival. The ECM acts as a reservoir of growth factors, cytokines, and matricellular proteins, which are secreted and stored within the matrix, allowing for controlled release and presentation to cells, directly influencing cell behavior through interactions with cell surface receptors, such as integrins, discoidin domain receptors [36-37](DDRs), and syndecans.Integrins, for example, are transmembrane receptors that mediate cell-ECM adhesion and transmit bidirectional signals between the cell and its environment, influencing intracellular signaling pathways like MAPK and Rho GTPases, which regulate cell motility and cytoskeletal organization [38]. Matricellular proteins like tenascins and osteopontin modulate cell-ECM interactions without directly contributing to the structural integrity of the matrix [36].
The composition of the ECM is extremely varied and specific to each type of tissue, reflecting its specialized functions and the dynamic requirements of the resident cells. In the dermis, the main components include:
Collagen: It is the major structural component of the dermal ECM, representing approximately 70-80% of the skin's dry weight, providing the skin with its tensile strength and structural integrity [39]. Collagen is synthesized as a procollagen precursor, which undergoes post-translational modifications, including hydroxylation of proline and lysine residues (requiring Vitamin C as a cofactor) and glycosylation, before being secreted into the extracellular space where it is cleaved by procollagen peptidases to form mature collagen molecules that self-assemble into fibrils and fibers.There are at least 28 different types of collagen, each encoded by a distinct gene. Type I collagen is predominant in the reticular dermis, providing resistance to stretching and being synthesized mainly by dermal fibroblasts.Other important types include type III collagen, which is more abundant in the papillary dermis and plays a role in the early stages of wound healing, providing a more pliable matrix that facilitates cell migration [40], type IV collagen, a major component of the basement membrane underlying the epidermis and surrounding blood vessels and type VII collagen, which forms anchoring fibrils that attach the basement membrane to the underlying dermis, ensuring epidermal-dermal cohesion [41]. Fibrillar collagens (types I, II, III, V, and XI) form the major structural framework of the ECM, while non-fibrillar collagens (e.g., types IV, VI, VII) have specialized functions in basement membranes, anchoring fibrils, and pericellular matrices. The regulation of collagen synthesis involves multiple factors, including growth factors (e.g., TGF-β, CTGF, PDGF), mechanical stimuli, and inflammatory mediators.
Elastin: Provides elasticity and the ability to stretch and recoil.. Elastic fibers form an interconnected three-dimensional network, giving skin its extensibility and retractability [42]. Elastin is synthesized primarily by fibroblasts in the dermis, as well as by smooth muscle cells in the walls of blood vessels, during development and early adulthood; elastin synthesis declines significantly with age [43]. Elastin monomers (tropoelastin) are secreted and assembled onto microfibrils composed of fibrillin-1, fibulin-5, and other proteins, forming elastic fibers. The cross-linking of tropoelastin molecules by lysyl oxidase (LOX) is essential for the formation of mature, functional elastin fibers [44]. The degradation of elastin, caused by factors such as chronic exposure to ultraviolet (UV) radiation from the sun and increased levels of proteolytic enzymes (matrix metalloproteinases - MMPs, particularly MMP-2 and MMP-9, and elastases) released by inflammatory cells like neutrophils and macrophages, contributes to the loss of skin elasticity (elastosis) and the formation of wrinkles associated with chronological aging and photoaging. Advanced glycation end-products (AGEs), formed through the non-enzymatic glycation of proteins, also contribute to elastin stiffening and reduced elasticity in aged skin [45].
Proteoglycans: These are complex macromolecules consisting of a core protein to which one or more glycosaminoglycan (GAG) chains are covalently attached. These GAGs, which are negatively charged due to the presence of sulfate and carboxyl groups, attract and retain water molecules, contributing to the hydration, turgor, and compressibility of the ECM [47]. Proteoglycans play a critical role in regulating ECM assembly, cell adhesion, growth factor signaling, and tissue homeostasis. Decorin, a small leucine-rich proteoglycan (SLRP), binds to collagen fibrils and regulates their diameter and organization, inhibiting excessive collagen deposition and fibrosis. Biglycan, another SLRP, also interacts with collagen and modulates cell signaling through Toll-like receptors (TLRs) [44]. Versican, a large aggregating proteoglycan, contributes to tissue hydration and cell migration during development and wound healing [48].
Glycosaminoglycans (GAGs): These are long, unbranched, linear polysaccharides composed of repeating disaccharide units. GAGs are highly negatively charged due to the presence of sulfate and uronic acid groups, and they have a high affinity for water. Major GAGs in the skin include hyaluronic acid (hyaluronan), chondroitin sulfate, dermatan sulfate, heparan sulfate, and keratan sulfate[49]. Hyaluronic acid (HA) is a non-sulfated GAG with an exceptional ability to bind and retain water, up to 1000 times its weight, significantly contributing to the hydration, viscoelasticity, compressibility, and mechanical properties of the dermal ECM. HA also plays a crucial role in cell migration, proliferation, and inflammation by interacting with cell surface receptors such as CD44 and receptor for hyaluronan-mediated motility (RHAMM). Chondroitin sulfate and dermatan sulfate are often found covalently attached to proteoglycans, influencing their interactions with other ECM components and regulating cell signaling [50]. GAGs, through their interaction with proteins and growth factors, modulate their activity and influence key cellular processes, including cell adhesion, migration, proliferation, and differentiation.
Adhesive Glycoproteins: These are a diverse group of multi-domain proteins that mediate cell-ECM and cell-cell interactions, playing a crucial role in cell adhesion, migration, differentiation, and tissue organization [51]. Fibronectin is a large, multifunctional glycoprotein that binds to a variety of ECM components, including collagen, fibrin, heparin, and heparan sulfate, as well as to cell surface integrin receptors, facilitating cell adhesion, migration, wound healing, and tissue remodeling. Fibronectin exists as a soluble dimer in plasma and as an insoluble multimer in the ECM, with different isoforms generated by alternative splicing of the fibronectin. Laminins are a family of heterotrimeric glycoproteins that are major components of basement membranes, playing a crucial role in the adhesion, migration, and differentiation of epithelial and endothelial cells, as well as in the structural organization and stability of the basement membrane. Laminins bind to cell surface integrins, dystroglycan, and other receptors, influencing cell signaling and tissue morphogenesis [52]. Other important adhesive glycoproteins include tenascins, thrombospondins, and vitronectin, each with distinct roles in regulating cell-ECM interactions and tissue remodeling.
We will also find new images in this section.
Figure 2. Extracellular Matrix (ECM): components and function (figure made by the authors)
Figure 3. Mechanisms of Fibroblast Response in Tissue Regeneration(figure made by the authors).
Comments 5: Section 4: also needs an image.
Sub-section of section 4 are too short. the authors should expand them by providing enough updated information from the previous studies.
Fig. 2 and Fig. 3 needs to be modified, suitable for a review paper.
Table 1, use number instead of Greek numbers. it needs citation if information extracted.
Also remove the blue color.
The MS needs a future perspective
Response 5: We thank you for the valuable suggestions. In response:
- We have added a new image in Section 4 to visually support the discussed content.
- The sub-sections in Section 4 have been expanded with updated information and relevant findings from recent literature.
- Figures 2 and 3 have been revised to ensure they are more suitable for a review article.
- Table 1 has been modified and appropriate citations have been included. The blue color has also been removed for consistency and clarity.
- Finally, a new section outlining future perspectives has been added at the end of the manuscript to highlight ongoing challenges and potential research directions.
We have substantially improved Section 4 by incorporating data from more recent studies. We have also added one illustrative figure.
Figure nr 4:DNA Damage and Cell Senescence: Pathways of Aging (figure made by the authors).
- Impact of Aging on Extracellular Matrix Synthesis and Remodeling: A Molecular and Cellular Perspective
These processes include changes in gene expression, protein synthesis, enzymatic activity, and cell-cell communication, all of which contribute to a decline in the structural integrity and functional capacity of the dermis. Furthermore, environmental factors such as UV radiation, pollution, and smoking can accelerate these age-related changes, leading to premature skin aging [63].
4.1. Diminished ECM Production
A key feature of skin aging is the decreased production of important components of the extracellular matrix. This decline impacts several essential structural proteins and glycosaminoglycans, which are vital for maintaining healthy, youthful skin:
Collagen: Specifically, the activity of prolyl hydroxylase and enzymes responsible for hydroxylating proline and lysine residues in collagen, decreases with age, resulting in reduced collagen stability and crosslinking [67]. The ratio of type I to type III collagen also changes with age, with a relative increase in type III collagen, which has a smaller fiber diameter and lower tensile strength [68], the expression of collagen chaperones, such as heat shock protein 47 (HSP47), which are essential for proper collagen folding and secretion, is reduced in aged fibroblasts [69]. Meanwhile, existing collagen fibers exhibit increased cross-linking, resulting in a less organized and more rigid structure with reduced tensile strength. These cross-links are formed by advanced glycation end products (AGEs) and enzymatic cross-linking mediated by lysyl oxidase (LOX) [54]. The accumulation of AGEs not only stiffens collagen fibers but also makes them more susceptible to degradation by matrix metalloproteinases (MMPs) [71]. This altered collagen structure contributes significantly to the reduced elasticity and increased susceptibility to wrinkles and sagging observed in aged skin.
Elastin: Breakdown and loss of elasticity: Elastin fibers, responsible for the elasticity and resilience of the dermis, also undergo significant age-related changes. The synthesis of tropoelastin, the precursor to elastin, diminishes with age, resulting in fewer elastin fibers and reduced overall elastic recoil ,The decline in tropoelastin synthesis is associated with decreased expression of the elastin gene and reduced stability of elastin mRNA [73]. In addition, age-related changes in the microfibrillar network, which provides a scaffold for elastin deposition, can impair the proper assembly and organization of elastic fibers [74]. Furthermore, existing elastin fibers become fragmented and cross-linked, leading to reduced elasticity and increased susceptibility to wrinkle formation [75]. Elastin fragmentation is primarily mediated by MMPs, particularly MMP-2, MMP-9, and MMP-12, which are upregulated in aged skin[76]. Cross-linking of elastin fibers by AGEs and LOX further reduces their elasticity and makes them more resistant to degradation [77]. The accumulation of damaged elastin fragments in the dermis, known as solar elastosis, is a hallmark of photoaged skin [78].This loss of elasticity contributes to the loss of skin turgor and the appearance of wrinkles and sagging.
Glycosaminoglycans (GAGs): Reduced hydration and modified biomechanical properties: GAGs, including hyaluronic acid, are essential for maintaining the hydration and viscoelasticity of the dermis [79]. Age-related declines in GAG synthesis lead to decreased water retention in the dermis, contributing to increased skin dryness and altered biomechanical properties. The synthesis of hyaluronic acid (HA) by HA synthases (HAS1, HAS2, and HAS3) is reduced in aged fibroblasts, leading to a decrease in HA content in the dermis [80]. In addition, the degradation of HA by hyaluronidases (HYAL1, HYAL2, and HYAL3) is increased with age, further contributing to the decline in HA levels [81]. The sulfation of other GAGs, such as chondroitin sulfate and dermatan sulfate, is also reduced in aged skin, affecting their interactions with other ECM components and their ability to regulate cell signaling [82]. This diminished hydration further contributes to the appearance of wrinkles and reduced skin resilience [83].
4.2 Enhanced ECM Degradation
The age-related decline in ECM production is highly exacerbated by higher degradation in ECM mediated by several key processes:
- Upregulation of Matrix Metalloproteinases (MMPs): The activity of various MMPs, especially MMP-1 (collagenase-1), MMP-2 (gelatinase A), and MMP-9 (gelatinase B), becomes more active with age [84]. The expression of MMPs is regulated by a variety of factors, including growth factors, cytokines, and UV radiation [85]. Aged fibroblasts exhibit increased expression of MMPs due to increased activity of transcription factors such as AP-1 and NF-κB [86]. Furthermore, the levels of reactive oxygen species (ROS) are elevated in aged skin, which can activate MMPs and promote ECM degradation [87]. This increased MMP activity, often exceeding the capacity of tissue inhibitors of metalloproteinases (TIMPs) to neutralize them, leads to the net degradation of collagen and elastin fibers [88].The imbalance between MMPs and TIMPs is a key factor in age-related ECM degradation . The expression of TIMPs, particularly TIMP-1 and TIMP-2, is reduced in aged fibroblasts, further contributing to the increased MMP activity [89]. The imbalance between MMP and TIMP activity makes the ECM structure less stable and accelerates aging phenotypes.
- Accumulation of Advanced Glycation End Products (AGEs): AGEs are formed through the non-enzymatic glycation of ECM proteins, particularly collagen and elastin. AGEs crosslink ECM molecules, increasing their rigidity and susceptibility to degradation [40].Glycation is a process in which reducing sugars, such as glucose and fructose, react with amino groups in proteins to form Schiff bases, which undergo further reactions to form irreversible AGEs .AGEs accumulate in the skin with age, particularly in long-lived proteins such as collagen and elastin [91].Furthermore, AGEs stimulate the production of pro-inflammatory cytokines and reactive oxygen species (ROS), further damaging the ECM.
- Chronic Low-Grade Inflammation: : Chronic low-grade inflammation is a hallmark of aging and significantly contributes to ECM degradation . Inflammation is characterized by elevated levels of pro-inflammatory cytokines, such as TNF-α, IL-6, and IL-1β, in the circulation and in tissues [92].Inflammatory mediators, such as cytokines and chemokines, promote the activity of MMPs and inhibit collagen synthesis [84],[93]. Cytokines such as TNF-α and IL-1β stimulate the expression of MMPs in fibroblasts and inhibit the synthesis of collagen by suppressing the expression of collagen genes and interfering with collagen processing.Furthermore, inflammatory mediators can activate immune cells, such as macrophages and neutrophils, which release proteases and ROS that further contribute to ECM damage.[94]This combination further accelerates the breakdown of the ECM.
4.3 Cellular and Microenvironmental Alterations
Age-related changes in the cellular microenvironment also play a major role in altered ECM synthesis and remodeling:
Senescent Fibroblasts and the Senescence-Associated Secretory Phenotype (SASP): The accumulation of senescent fibroblasts in the dermis is a key feature of aging .Cellular senescence is a state of irreversible growth arrest characterized by morphological and functional changes . Senescent cells accumulate in tissues with age due to DNA damage, telomere shortening, oxidative stress, and oncogene activation [95]. These senescent fibroblasts release a complex mixture of factors, known as the SASP, which includes pro-inflammatory cytokines, growth factors, and MMPs. The SASP is a complex and dynamic secretome that includes a variety of factors, such as IL-6, IL-8, MMP-1, MMP-3, and VEGF .The SASP contributes to chronic inflammation and further promotes ECM degradation [96]. The SASP can also induce senescence in neighboring cells, leading to a self-perpetuating cycle of senescence and inflammation [95].
- Impaired Cell-Cell Interactions: The interactions between fibroblasts and other dermal cells, such as keratinocytes, become less efficient with age. Fibroblasts and keratinocytes communicate with each other through direct cell-cell contact and through the release of soluble factors, such as growth factors and cytokines .These altered interactions may result in impaired communication and coordination of ECM production and remodeling [97].Age-related changes in the expression of cell adhesion molecules, such as integrins and cadherins, can disrupt cell-cell interactions and impair the ability of fibroblasts and keratinocytes to communicate with each other. Furthermore, the decline in growth factor signaling in aged skin can impair the ability of keratinocytes to stimulate collagen synthesis in fibroblasts [98].
- Altered Growth Factor Signaling: The responsiveness of fibroblasts to various growth factors, such as TGF-β and FGF, is changed with age[99]. Growth factors play a critical role in regulating ECM synthesis and remodeling. TGF-β stimulates collagen synthesis and inhibits MMP expression in fibroblasts, while FGF stimulates fibroblast proliferation and migration [98]. These changes reduce the ability of fibroblasts to produce and remodel the ECM effectively.
4.4. Clinical Consequences
The age-related changes in ECM structure and function have several clinical consequences, including:
Increased wrinkle formation and reduced skin elasticity: The diminished collagen and elastin content, coupled with increased ECM degradation, directly leads to the visible signs of aging such as fine lines and deeper wrinkles [100]. The loss of elasticity reduces the skin's ability to recoil after stretching, resulting in persistent wrinkles and a less firm appearance. Furthermore, alterations in the organization of collagen fibers disrupt the skin's smooth surface, contributing to the formation of wrinkles and textural irregularities [101].
Skin sagging and loss of turgor: The reduced collagen and elastin support, along with decreased GAG-mediated hydration, results in a loss of skin volume and a decrease in its ability to resist gravity. This leads to sagging, particularly in areas like the cheeks, jawline, and under the eyes. The decreased turgor, or skin fullness, makes the skin appear thinner and more fragile[102]. Changes in the subcutaneous fat distribution, which also occur with aging, further contribute to the loss of facial volume and sagging .
Increased skin dryness and roughness: The decline in GAGs, especially hyaluronic acid, reduces the skin's ability to retain moisture, leading to increased dryness and a rough, uneven texture. This dryness can exacerbate the appearance of wrinkles and fine lines and can compromise the skin's barrier function, making it more susceptible to irritants and allergens. The altered lipid composition of the stratum corneum, which also occurs with aging, further contributes to the increased skin dryness.
Impaired wound healing: The age-related decline in fibroblast function and ECM remodeling capacity impairs the skin's ability to heal wounds effectively. Aged fibroblasts exhibit reduced proliferation and migration, and their ability to synthesize new collagen and other ECM components is compromised. The increased levels of MMPs in aged skin can also disrupt the formation of a stable wound matrix, leading to delayed wound closure and increased risk of scarring [103]. Furthermore, the reduced vascularity in aged skin can impair oxygen and nutrient delivery to the wound site, further delaying the healing process.
Increased susceptibility to skin damage and infections: The thinner, less elastic, and more fragile skin is more vulnerable to damage from external factors such as UV radiation, mechanical trauma, and chemical irritants [102]. The compromised skin barrier function makes it easier for pathogens to penetrate the skin, increasing the risk of infections .The reduced immune function in aged skin further contributes to the increased susceptibility to infections and delayed wound healing [104].
The table has been revised to incorporate bibliographic references and a modified numbering system.
|
Category |
Medications |
Mechanism of Action |
Target Cell Types |
Efficacy |
References |
|
Senolytics |
Dasatinib, Quercetin |
Inducing apoptosis in senescent cells |
Senescent cells from various tissues |
Significant reduction of sarcopenia |
[113] |
|
|
Navitoclax |
Blocking BCL-2 signaling to promote apoptosis |
Senescent cells, especially from adipose tissue |
Improvement of muscle function |
[114,115] |
|
Geroprotectors |
Rapamycin, Metformin |
Inhibiting the mTOR pathway to improve longevity |
Cells from different types of tissue |
Delaying the aging process |
[116,117] |
|
|
Curcumin |
Activating the Nrf2 pathway to reduce oxidative stress |
Cells in general, especially those involved in inflammation |
Reducing inflammation and oxidative stress |
[118-120] |
|
|
Resveratrol |
Activating the SIRT1 protein to induce longevity |
Cells from all tissues, including muscular and nervous |
Improving metabolism and cardiovascular function |
[121,122] |
- Orioli, D.; Dellambra, E. Epigenetic Regulation of Skin Cells in Natural Aging and Premature Aging Diseases. Cells 2018, 7, 268, doi:10.3390/cells7120268.
- Zheng, L.; He, S.; Wang, H.; Li, J.; Liu, Y.; Liu, S. Targeting Cellular Senescence in Aging and Age-Related Diseases: Challenges, Considerations, and the Emerging Role of Senolytic and Senomorphic Therapies. Aging Dis 2024, 15, 2554–2594, doi:10.14336/AD.2024.0206.
- Zhu, Y.; Tchkonia, T.; Fuhrmann-Stroissnigg, H.; Dai, H.M.; Ling, Y.Y.; Stout, M.B.; Pirtskhalava, T.; Giorgadze, N.; Johnson, K.O.; Giles, C.B.; et al. Identification of a Novel Senolytic Agent, Navitoclax, Targeting the Bcl-2 Family of Anti-Apoptotic Factors. Aging Cell 2016, 15, 428–435, doi:10.1111/acel.12445.
- Elliehausen, C.J.; Anderson, R.M.; Diffee, G.M.; Rhoads, T.W.; Lamming, D.W.; Hornberger, T.A.; Konopka, A.R. Geroprotector Drugs and Exercise: Friends or Foes on Healthy Longevity? BMC Biol 2023, 21, 287, doi:10.1186/s12915-023-01779-9.
- Le Couteur, D.G.; Barzilai, N. New Horizons in Life Extension, Healthspan Extension and Exceptional Longevity. Age Ageing 2022, 51, afac156, doi:10.1093/ageing/afac156.
- Ashrafizadeh, M.; Ahmadi, Z.; Mohammadinejad, R.; Farkhondeh, T.; Samarghandian, S. Curcumin Activates the Nrf2 Pathway and Induces Cellular Protection Against Oxidative Injury. Curr Mol Med 2020, 20, 116–133, doi:10.2174/1566524019666191016150757.
- Cui, J.; Li, H.; Zhang, T.; Lin, F.; Chen, M.; Zhang, G.; Feng, Z. Research Progress on the Mechanism of Curcumin Anti-Oxidative Stress Based on Signaling Pathway. Front. Pharmacol. 2025, 16, doi:10.3389/fphar.2025.1548073.
- D, L.; D, S.; Y, W.; Y, C.; N, L.; J, T.; D, Y.; Y, S. Curcumin Protects against Doxorubicin Induced Oxidative Stress by Regulating the Keap1-Nrf2-ARE and Autophagy Signaling Pathways. Psychopharmacology 2023, 240, doi:10.1007/s00213-023-06357-z.
- Zhang, L.; Pitcher, L.E.; Prahalad, V.; Niedernhofer, L.J.; Robbins, P.D. Recent Advances in the Discovery of Senolytics. Mech Ageing Dev 2021, 200, 111587, doi:10.1016/j.mad.2021.111587.
- Blagosklonny, M.V. Rapamycin for Longevity: Opinion Article. Aging (Albany NY) 2019, 11, 8048–8067, doi:10.18632/aging.102355.
In the end we made also a new paragraph with future perspectives of our article
Future studies should focus on translating the promising preclinical findings of senolytics and geroprotectors into effective and safe human therapies for skin aging, with particular attention to personalized approaches that account for individual variations in aging mechanisms and responses.And further research is needed to elucidate the intricate interplay between dermal fibroblast plasticity, cellular senescence, and ECM remodeling in the context of skin aging, particularly in identifying novel molecular targets and biomarkers that can predict and monitor the effectiveness of interventions.

Round 2
Reviewer 1 Report
Comments and Suggestions for Authors
The manuscript has been significantly improved by the authors' complete reworking of the bibliography list, significant additions to the text, and editing figures.
However, the latest version has a few incorrectly formatted references (67, 69, 70, 71, 73, 76, 77, 120, etc. - Authors) and there are other technical errors in the text.
Author Response
Comments 1: The manuscript has been significantly improved by the authors' complete reworking of the bibliography list, significant additions to the text, and editing figures.
However, the latest version has a few incorrectly formatted references (67, 69, 70, 71, 73, 76, 77, 120, etc. - Authors) and there are other technical errors in the text.
Response 1: We sincerely thank you for the constructive feedback and appreciation of the substantial improvements made to the manuscript. In response to the remaining concerns:
- We have carefully reviewed and corrected the formatting of the references indicated (67, 69, 70, 71, 73, 76, 77, 120)
- We have standardized the formatting throughout the manuscript and clearly distinguished all subheadings to enhance the overall structure.
- We have inserted clear references in the text to Figures 1, 3, and 4 to enhance clarity and guidance for the reader.
- Table 1 has been reformatted using the MDPI template, ensuring alignment with the journal's publication standards.
We hope these changes adequately address the reviewer’s comments. Thank you again for your valuable input.
Reviewer 2 Report
Comments and Suggestions for Authors
The authors answered all my requests; therefore, the manuscript can be accepted.
Author Response
Comments 1: The authors answered all my requests; therefore, the manuscript can be accepted.
Response 1: Thank you very much for your positive feedback and for the time and effort you dedicated to reviewing our manuscript. We appreciate your helpful comments throughout the revision process.
Reviewer 3 Report
Comments and Suggestions for Authors
the authors addressed my comments well.
Author Response
Comments 1: the authors addressed my comments well.
Response 1: Thank you very much for your positive feedback and for the time and effort you dedicated to reviewing our manuscript. We appreciate your helpful comments throughout the revision process.